# PoGDiff: Product-of-Gaussians Diffusion Models for Imbalanced Text-to-Image Generation

## Abstract

Diffusion models have made significant advancements in recent years. However, their performance often deteriorates when trained or fine-tuned on imbalanced datasets. This degradation is largely due to the disproportionate representation of majority and minority data in image-text pairs. In this paper, we propose a general fine-tuning approach, dubbed PoGDiff, to address this challenge. Rather than directly minimizing the KL divergence between the predicted and ground-truth distributions, PoGDiff replaces the ground-truth distribution with a Product of Gaussians (PoG), which is constructed by combining the original ground-truth targets with the predicted distribution conditioned on a neighboring text embedding. Experiments on real-world datasets demonstrate that our method effectively addresses the imbalance problem in diffusion models, improving both generation accuracy and quality.

## 1 Introduction

The development of diffusion models (Ho et al., 2020; Song et al., 2020b) and their subsequent extensions (Song et al., 2020a; Nichol & Dhariwal, 2021; Huang et al., 2023) has significantly advanced the learning of complex probability distributions across various data types, including images (Ho et al., 2022; Rombach et al., 2022; Saharia et al., 2022; Ho & Salimans, 2022), audio (Kong et al., 2020), and 3D biomedical imaging data (Luo & Hu, 2021; Poole et al., 2022; Shi et al., 2023; Pinaya et al., 2022). For these generative models, the amount of training data plays a critical role in determining both the accuracy of probability estimation and the model's ability to generalize, which enables effective extrapolation within the probability space.

Data diversity and abundance are key to improving the generative capabilities of large-scale models, enabling them to capture intricate details within a vast probability space. However, many data-driven modeling tasks often rely on small, imbalanced real-world datasets, leading to poor generation quality, particularly for minority groups. For example, when training and fine-tuning a diffusion model with an imbalanced dataset of individuals, existing models often struggle to generate accurate images for those who appear less frequently (i.e., minorities) in the training data (Fig. 1).

This limitation is true even for finetuning large diffusion models pretrained on large-scale datasets like LAION-5B (Schuhmann et al., 2022), e.g., Stable Diffusion (Rombach et al., 2022). Imagine an imbalanced dataset consisting of employees in a small company, senior employees might have more photos available, while new employees only have a very limited number of them. Since none of the employees appear in the LAION-5B dataset, generating photos of them require finetuning the Stable Diffusion model. Unfortunately, finetuning the model on such an imbalanced dataset might enable the model to generate accurate images for the majority group (i.e., senior employees), but it will perform poorly for the minority group (i.e., new employees).

To address this challenge, we propose a general fine-tuning approach, dubbed PoGDiff. Rather than directly minimizing the KL divergence between the predicted and ground-truth distributions, PoGDiff replaces the ground-truth distribution with a Product of Gaussians (PoG), which is constructed by combining the original ground-truth targets with the predicted distribution conditioned on a neighboring text embedding. Our contributions are as follows:

Figure 1: **PoGDiff for imbalanced text-to-image (IT2I) generation.** Existing methods such as Stable Diffusion (Rombach et al., 2022) and CBDM (Qin et al., 2023) fall short for minority data (**Low Density**). In contrast, Our PoGDiff successfully generates high-quality images even for minority data, outperforming all the baselines.

- We identify the problem of imbalanced text-to-image generation (IT2I) and introduce the first general diffusion model, dubbed Product-of-Gaussians Diffusion Models (PoGDiff), for addressing this problem.
- Our theoretical analysis shows that training of PoGDiff is equivalent to training a normal diffusion model while encouraging the model to generate the same image given similar text prompts (conditions).
- Our empirical results on real-world datasets demonstrate the effectiveness of our method, outperforming all state-of-the-art baselines.

## 2 RELATED WORK

**Long-Tailed Recognition.** Addressing the challenges posed by long-tailed data distributions has been a critical area of research in machine learning, for both classification and regression problems. Traditional methods, such as re-sampling and re-weighting techniques, have been used to mitigate class imbalances by either over-sampling minority classes or assigning higher weights to them during training (Chawla et al., 2002; He & Garcia, 2009; Torgo et al., 2013; Branco et al., 2017; 2018). Such algorithms fail to measure the distance in continuous label space and fall short in handling high-dimensional data (e.g., images and text). Deep imbalanced regression methods (Yang et al., 2021; Ren et al., 2022; Gong et al., 2022; Keramati et al., 2023; Wang & Wang, 2024) address this challenge by reweighting the data using the effective label density during representation learning. However, all these methods above are designed for *recognition* tasks such as classification and regression, and are therefore not applicable to our *generation* task.

**Diffusion Models Related to Long-Tailed Data.** There are also works that related to both diffusion models and long-tailed data. They aim at improving generation robustness using noisy label (Na et al., 2024), improving fairness in image generation (Shen et al., 2023), and improving classification accuracy using diffusion models (Zhang et al., 2024). However, these works have different goals and therefore are not applicable to our setting.

Most relevant to our work is Class Balancing Diffusion Model (CBDM) (Qin et al., 2023), which uses a distribution adjustment regularizer that enhances tail-class generation based on the model's predictions for the head class. It improves the quality of long-tailed generation by assuming one-hot conditional labels (i.e., classification-based settings). However, this assumption does not generalize to the modern setting where image generation is usually conditioned on free-form text prompts. As a result, when adapted to the free-form setting, they often fail to model the similarity among different text prompts, leading to suboptimal generation performance in minority data (as verified by empirical results in Sec. 4).

## 3 METHODS

### 3.1 PRELIMINARIES

**Diffusion models (DMs)** (Ho et al., 2020) are probabilistic models that generate an output image $\mathbf{x}_0$ from a random noise vector $\mathbf{x}_T$ conditioned on text input $\mathbf{c}$. DMs operate through two main processes: the forward diffusion process and the reverse denoising process. During the diffusion process, Gaussian noise is progressively added to a data sample $\mathbf{x}_0$ over $T$ steps. The forward process is defined as a Markov chain, where:

$$q\left(\mathbf{x}_t|\mathbf{x}_{t-1}\right) = \mathcal{N}\left(\mathbf{x}_t; \sqrt{1-\beta_t}\mathbf{x}_{t-1}, \beta_t \mathbf{I}\right).$$

Here, $\beta_t$ is the predefined diffusion rate at step $t$. By denoting $\alpha_t = 1 - \beta_t$ and $\bar{\alpha}_t = \prod_{i=1}^{t}\alpha_i$, we can describe the entire diffusion process as:

$$q\left(\mathbf{x}_{1:T}|\mathbf{x}_0\right) = \prod_{t=1}^{T} q\left(\mathbf{x}_t|\mathbf{x}_{t-1}\right)$$
$$q\left(\mathbf{x}_t|\mathbf{x}_0\right) = \mathcal{N}\left(\mathbf{x}_t; \sqrt{\bar{\alpha}_t}\mathbf{x}_0, (1-\bar{\alpha}_t)\mathbf{I}\right)$$

The denoising process removes noise from the sample $\mathbf{x}_T$, eventually recovering $\mathbf{x}_0$. A denoising model $\epsilon_\theta(\mathbf{x}_t, t, \mathbf{y})$ is trained to estimate the noise $\epsilon$ from $\mathbf{x}_t$ and a text-guided embedding $\mathbf{y} = \phi(\mathbf{c})$, where $\phi(\cdot)$ is a pretrained text encoder. Formally:

$$p_\theta\left(\mathbf{x}_{t-1}|\mathbf{x}_t, t, \mathbf{y}\right) = \mathcal{N}\left(\mathbf{x}_{t-1}; \epsilon_\theta(\mathbf{x}_t, t, \mathbf{y}), \sigma_t^2\mathbf{I}\right).$$

The denoising process is trained by maximizing the likelihood of the data under the model or, equivalently, by minimizing the variational lower bound on the negative log-likelihood of the data. Ho et al. (2020) shows that this is equivalent to minimizing the KL divergence between the predicted distribution $p_\theta(\mathbf{x}_{t-1}|\mathbf{x}_t, \mathbf{y})$ and the ground-truth distribution $q(\mathbf{x}_{t-1}|\mathbf{x}_t, \mathbf{x}_0, \mathbf{y})$ at each time step $t$ during the backward process. The training objective then becomes:

$$\min D_{KL}\left(q\left(\mathbf{x}_{t-1}|\mathbf{x}_t, \mathbf{x}_0, \mathbf{y}\right) \middle\| p_\theta\left(\mathbf{x}_{t-1}|\mathbf{x}_t, \mathbf{y}\right)\right).$$

This can be simplified to:

$$L_{DM} = \mathbb{E}_{\mathbf{x}_0=\mathbf{x}, \epsilon\sim\mathcal{N}(0,\mathbf{I}),t}\left[\|\epsilon - \epsilon_\theta(\mathbf{x}_t, t, \mathbf{y})\|_2^2\right].$$

**Latent diffusion models (LDMs)** (Rombach et al., 2022) are diffusion models that perform the entire diffusion and denoising process in a lower-dimensional latent space. LDMs first learn an encoder $\mathcal{E}$ and a decoder $\mathcal{D}$, which are then frozen during subsequent training of the diffusion models. The corresponding objective is then simplified to:

$$L_{LDM} = \mathbb{E}_{\mathbf{z}_0=\mathcal{E}(\mathbf{x}), \epsilon\sim\mathcal{N}(0,\mathbf{I}),t}\left[\|\epsilon - \epsilon_\theta(\mathbf{z}_t, t, \mathbf{y})\|_2^2\right]$$

In this paper, we use Stable Diffusion (LDM) (Rombach et al., 2022) as our backbone model. Since our method works for both the vanilla DMs and LDMs, for clarity, we use the notation $\mathbf{x}$ instead of $\mathbf{z}$, as the encoder $\mathcal{E}$ and decoder $\mathcal{D}$ are fixed during fine-tuning.

### 3.2 PRODUCT-OF-GAUSSIANS DIFFUSION MODELS (POGDIFF)

#### 3.2.1 MAIN IDEA

**Method Overview.** Given an image dataset $\mathcal{D} = \{\mathbf{x}^{(i)}, \mathbf{c}^{(i)}\}_{i=1}^{N}$, where $\mathbf{c}^{(i)}$ is the text description for image $\mathbf{x}^{(i)}$, we use a fixed CLIP encoder to produce $\mathbf{c}^{(i)}$'s corresponding text embedding $\mathbf{y} = \phi(\mathbf{c})$.

Typical diffusion models minimize the KL divergence between the predicted distribution $p_\theta(\mathbf{x}_{t-1}|\mathbf{x}_t, \mathbf{y}) = \mathcal{N}(\epsilon_\theta(\mathbf{x}_t, t, \mathbf{y}), \lambda_{\mathbf{y}}^{-1}\mathbf{I})$ and the ground-truth distribution $q(\mathbf{x}_{t-1}|\mathbf{x}_t, \mathbf{x}_0, \mathbf{y}) = \mathcal{N}(\epsilon, \lambda_t^{-1}\mathbf{I})$ at each

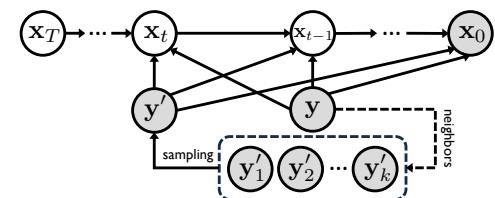

Figure 2: Overview of our PoGDiff. During fine-tuning, PoGDiff collects $k$ neighbors of the current text embedding $\mathbf{y}$ and samples one $\mathbf{y}'$ from them based on Eqn. (8). Both $\mathbf{y}$ and $\mathbf{y}'$ will then be employed to denoise the current image $\mathbf{x}_t$ to $\mathbf{x}_{t-1}$.

time step $t$ during the backward denoising process. Here, $\lambda_{\mathbf{y}}$ and $\lambda_t$ represent the precision. In contrast, our PoGDiff replaces the ground-truth target with a Product of Gaussians (PoG), and instead minimize the following KL divergence (for each $t$)

$$\mathcal{L}_{t-1}^{\text{PoGDiff}} = D_{KL}\left(q\left(\mathbf{x}_{t-1}|\mathbf{x}_t, \mathbf{x}_0, \mathbf{y}\right) \circ p_\theta\left(\mathbf{x}_{t-1}|\mathbf{x}_t, \mathbf{y}'\right) \middle\| p_\theta\left(\mathbf{x}_{t-1}|\mathbf{x}_t, \mathbf{y}\right)\right), \tag{1}$$

where $\circ$ represents the product of two Gaussian distributions, $\mathbf{y}'$ is a selected neighboring embedding from other samples in the training dataset (more details below), and $p_\theta(\mathbf{x}_{t-1}|\mathbf{x}_t, \mathbf{y}')$ denotes the predicted distribution when using $\mathbf{y}'$ as the input text embedding.

As shown in Fig. 2, intuitively, PoGDiff's denoising model $\epsilon_\theta(\mathbf{x}_t, t, \mathbf{y})$ (or $p_\theta(\mathbf{x}_{t-1}|\mathbf{x}_t, \mathbf{y})$) is optimized towards two target distributions, equivalently increasing the weights for minority instances (more details below). This approach enhances the text-to-image mapping by leveraging the statistical strength of neighboring data points, thereby improving and quality of the generated images, especially for minority images.

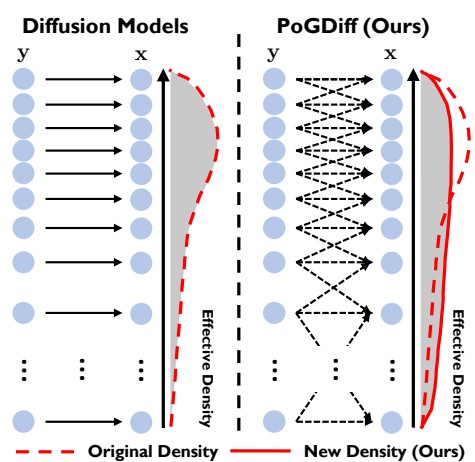

**Intuition behind the Product of Gaussians (PoG).** During fine-tuning, typical diffusion models "lock" the text conditional embedding $\mathbf{y} = \psi(c)$ to the corresponding image $\mathbf{x}$. Consequently, if the dataset follows a long-tailed distribution, the fine-tuned or post-trained diffusion model becomes heavily biased toward the majority data, performing poorly on minority data. Fig. 3 demonstrates our intuition. When training using a text-image pair $(\mathbf{y}, \mathbf{x})$, our PoGDiff "borrows" information from neighboring text conditional embedding $\mathbf{y}'$, thereby effectively increasing the data density in the minority region and leading to smoother (less imbalanced) effective density, as shown in Fig. 3 (right). However, since the text embedding is fixed during fine-tuning (i.e., $\phi$ is frozen), directly smoothing the text embedding space is not feasible. Instead, we rely on the properties of the product of Gaussian distributions.

Figure 3: Comparing denoising networks of typical diffusion models (Ho et al., 2020; Rombach et al., 2022) and our PoGDiff. **Left:** In conditional text-to-image diffusion models, a data point (i.e., $\mathbf{x}$) is mainly affected by its text embedding (also affected by the random latent codes). **Right:** In PoGDiff, neighbors participate to modulate the final effective density. Here, $\mathbf{y}$ denotes the text prompts, which are the embeddings of the text descriptions of the images; $\mathbf{x}$ denotes the associated images. The tightly packed circles at the top indicate higher density, while the sparse circles indicate lower density.

By definition, given two Gaussian distributions, $\mathcal{N}(\mu_1, \lambda_1^{-1}\mathbf{I})$ and $\mathcal{N}(\mu_2, \lambda_2^{-1}\mathbf{I})$, their product is still a Gaussian distribution:

$$\mathcal{N}(\mu_1, \lambda_1^{-1}) \circ \mathcal{N}(\mu_2, \lambda_2^{-1}) = \mathcal{N}\left(\frac{\lambda_1\mu_1 + \lambda_2\mu_2}{\lambda_1 + \lambda_2}, (\lambda_1 + \lambda_2)^{-1}\right) \triangleq \mathcal{N}\left(\mu_{\text{PoG}}, \lambda_{\text{PoG}}^{-1}\right), \tag{2}$$

which can be treated as a "composition" of two individual Gaussians, incorporating information from both. This intuition is key to developing our PoGDiff objective function.

### 3.2.2 THEORETICAL ANALYSIS AND ALGORITHMIC DESIGN

Based on Eqn. (1), we then derive a concrete objective function following Proposition 3.1 below.

**Proposition 3.1.** *Assume $\lambda_{\mathbf{y}} = \lambda_{PoG} \triangleq \lambda_t + \lambda_{\mathbf{y}'}$, we have our loss function*

$$\mathcal{L}_{t-1}^{PoGDiff} = \mathbb{E}_q\left[\frac{\lambda_{\mathbf{y}}}{2}\|\mu_\theta(\mathbf{x}_t, \mathbf{y}) - \mu_{PoG}\|^2\right] + C. \tag{3}$$

*Here, $C$ is a constant, and $\mu_{PoG}$ denotes the mean of the PoG, with the expression defined in Eqn. (2). Then, through derivations based on Gaussian properties, we obtain*

$$\mathcal{L}_{t-1}^{PoGDiff} \leq \mathbb{E}_q\left[\mathcal{A}(\lambda_t)\|\epsilon_\theta(\mathbf{x}_t, \mathbf{y}) - \epsilon\|^2 + \mathcal{A}(\lambda_{\mathbf{y}'})\|\epsilon_\theta(\mathbf{x}_t, \mathbf{y}) - \epsilon_\theta(\mathbf{x}_t, \mathbf{y}')\|^2\right] + C \tag{4}$$

*where the function $\mathcal{A}(\lambda) \triangleq \frac{\lambda(1-\alpha_t)^2}{2\alpha_t(1-\bar{\alpha}_t)}$.*

The proof is available in the Appendix. Eqn. (4) in Proposition 3.1 provides a upper bound for the KL divergence (Eqn. (1)) we aim to minimize.

In diffusion model literature (Ho et al., 2020; Rombach et al., 2022), one typically sets $\mathcal{A}(\lambda_t) = 1$ to eliminate the dependency on the time step $t$, and thus Eqn. (4) can be written as[1]:

$$\mathcal{L}_{\text{simple}}^{\text{PoGDiff}} = \mathbb{E}_{\mathbf{x}_0 \sim q(\mathbf{x}_0), \epsilon \sim \mathcal{N}(\mathbf{0}, \mathbf{I}), t \sim \mathcal{U}(1, T)} \left[ \|\epsilon_\theta(\mathbf{x}_t, \mathbf{y}) - \epsilon\|^2 + \frac{\lambda_{\mathbf{y}'}}{\lambda_t} \|\epsilon_\theta(\mathbf{x}_t, \mathbf{y}) - \epsilon_\theta(\mathbf{x}_t, \mathbf{y}')\|^2 \right]. \quad (5)$$

For convenience, we rewrite $\frac{\lambda_{\mathbf{y}'}}{\lambda_t} = \frac{\sigma_t^2}{\sigma_{\mathbf{y}'}^2}$. Note that this weight still depends on the time step $t$. Therefore, to be consistent with the DDPM-related literature (Ho et al., 2020; Rombach et al., 2022), we hypothetically define $\sigma_{\mathbf{y}'}^2 = \frac{\sigma_t^2}{\psi\left[(\mathbf{x}, \mathbf{y}), (\mathbf{x}', \mathbf{y}')\right]}$ to cancel out the term $\sigma_t^2$, thereby effectively removing the time step dependency; here $\psi\left[(\mathbf{x}, \mathbf{y}), (\mathbf{x}', \mathbf{y}')\right]$ denotes the similarity between the two image-text pairs. By shortening the notation $\psi\left[(\mathbf{x}, \mathbf{y}), (\mathbf{x}', \mathbf{y}')\right]$ to $\psi$, we can further rewrite the objective function for PoGDiff as:

$$\mathcal{L}_{\text{simple}}^{\text{PoGDiff}} = \mathbb{E}_{\mathbf{x}_0 \sim q(\mathbf{x}_0), \epsilon \sim \mathcal{N}(\mathbf{0}, \mathbf{I}), t \sim \mathcal{U}(1, T)} \left[ \|\epsilon_\theta(\mathbf{x}_t, \mathbf{y}) - \epsilon\|^2 + \psi \|\epsilon_\theta(\mathbf{x}_t, \mathbf{y}) - \epsilon_\theta(\mathbf{x}_t, \mathbf{y}')\|^2 \right]. \quad (6)$$

### 3.2.3 COMPUTING THE SIMILARITY $\psi$

Next, we discuss the choice of $\psi$ in Eqn. (6). Given a image-text dataset $\mathcal{D}$, the similarities between each image-text pair need to be considered in two parts:

$$\psi \triangleq \psi_{\text{img-sim}}(\mathbf{x}, \mathbf{x}') \cdot \psi_{\text{inv-txt-den}}(\mathbf{y}), \quad (7)$$

where $\psi_{\text{img-sim}}(\mathbf{x}, \mathbf{x}')$ is the similarity between images $\mathbf{x}$ and $\mathbf{x}'$, and $\psi_{\text{inv-txt-den}}(\mathbf{y})$ is the probability density of the text embedding $\mathbf{y}$ (more details below).

**Image Similarity $\psi_{\text{img-sim}}$.** For all $\mathbf{x} \sim \mathcal{D}$, we apply a pre-trained image encoder to obtain the latent representations $\mathbf{z}$. We then calculate the cosine similarities between each $\mathbf{z}$ and select the $k$ nearest neighbors with the highest similarity values for all samples in the dataset $\mathcal{D}$, denoted as $[s_j]_{j=1}^k$, where $s_j$ represents the cosine similarity scores between $\mathbf{x}$ and other images in $\mathcal{D}$, sorted in descending order. These values are then normalized to produce the weights for each neighbor:

$$w_j = \frac{s_j}{\sum_j s_j}, \quad (8)$$

For each data pair $(\mathbf{x}, \mathbf{y})$, we then randomly sample a neighboring pair $(\mathbf{x}', \mathbf{y}')$ through from a categorical distribution $Cat([w_j]_{j=1}^k)$[2], i.e., with $w_j$ serving as the probability weight, and compute their image similarity as:

$$\psi_{\text{img-sim}}(\mathbf{x}, \mathbf{x}') \triangleq \max\left(0, s^{a_1 + a_2 \cdot \mathbb{1}\left[\mathcal{I}(\mathbf{x}) \neq \mathcal{I}(\mathbf{x}')\right]}\right), \quad (9)$$

where $s$ denotes the cosine similarity sampled by the weights $\{w_j\}$ defined in Eqn. (8), $\mathbb{1}[\cdot]$ denotes the indicator function, and $\mathcal{I}(\cdot)$ retrieves the class/identity of the current input image; for example, $\mathbb{1}[\mathcal{I}(\mathbf{x}) \neq \mathcal{I}(\mathbf{x}')] = 0$ if $\mathbf{x}$ and $\mathbf{x}'$ are two photos of the same person (e.g., Albert Einstein), and $\mathbb{1}[\mathcal{I}(\mathbf{x}) \neq \mathcal{I}(\mathbf{x}')] = 1$ if $\mathbf{x}$ and $\mathbf{x}'$ are photos of two different persons (e.g., $\mathbf{x}$ is Einstein and $\mathbf{x}'$ Reagan). $a_1, a_2$ are hyperparameters that control the scale of the similarities.[3] The intuition is to compute the image similarity according to both the image content similarity, i.e., $s$, and identity similarity, i.e., $\mathcal{I}(\mathbf{x})$ and $\mathcal{I}(\mathbf{x}')$.

---

[1]For clarification, our $\mathcal{A}(\lambda_t)$ is equivalent to $\lambda_t$ in (Ho et al., 2020), with the difference that in our paper, $\lambda$ refers to the precision of the Gaussian distribution.

[2]The term "Cat" refers to a "Categorical" distribution. For example, $Cat([0.2, 0.5, 0.3])$ represents a three-dimensional categorical distribution, where there is a 0.2 probability of selecting the first category, 0.5 probability of selecting the second, and 0.3 probability of selecting the third.

[3]For example, if the cosine similarity ($s$) between $x$ and $x'$ is 0.4, and $a_1 = a_2 = 1$: if $x$ and $x'$ are of the same person, the image similarity will be $0.4^1$, whereas if $x$ and $x'$ are not of the same person, the image similarity will be $0.4^2$, which is smaller.

---

**Algorithm 1** Training Algorithm for PoGDiff

---

1: **Inputs:** A dataset $\mathcal{D} = \left\{ \mathbf{x}^{(i)}, \mathbf{c}^{(i)} \right\}_{i=1}^{N}$.
2: **repeat**
3:     $(\mathbf{x}_0, \mathbf{c}) \sim \mathcal{D}$
4:     $\mathbf{y} = \phi(\mathbf{c})$
5:     $t \sim \text{Uniform}(1, \cdots, T)$
6:     $\epsilon \sim \mathcal{N}(\mathbf{0}, \mathbf{I})$
7:     Generate $\mathbf{y}'$ and $\psi$ from Eqn. (12)
8:     Calculate $\mathbf{x}_t = \sqrt{\bar{\alpha}_t}\mathbf{x}_0 + \sqrt{1 - \bar{\alpha}_t}\epsilon$
9:     Take gradient descent step on
10:       $\nabla_\theta \left[ \|\epsilon - \epsilon_\theta(\mathbf{x}_t, \mathbf{y})\|_2^2 + \psi\|\epsilon_\theta(\mathbf{x}_t, \mathbf{y}') - \epsilon_\theta(\mathbf{x}_t, \mathbf{y})\|_2^2 \right]$
11: **until** converged

---

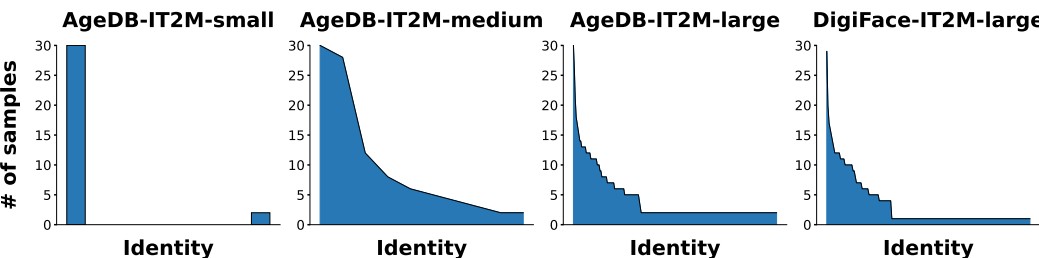

Figure 4: Overview of label distribution for four IT2I datasets. The x-axis corresponds to the identities (i.e., people or individuals).

**Inverse Text Densities** $\psi_{\text{inv-txt-den}}$. Inspired by LDS in DIR (Yang et al., 2021) and the theoretical analysis in VIR (Wang & Wang, 2024), re-weighting the label distribution of an imbalanced dataset can increase the optimization scale for minority classes and reduce the emphasis on majority classes, resulting in better performance under imbalanced conditions. However, both DIR and VIR partition the label space into bins, treating it as a classification problem. This is *not applicable* to our setting because in text-to-image generation, the "label" is actually text embeddings. Instead, we train a variational autoencoder (VAE) on this dataset and then approximate its likelihood $p(\mathbf{y})$ through its evidence lower bound, or ELBO:

$$p(\mathbf{y}) = e^{\log p(\mathbf{y})} \approx e^{\text{ELBO}_{\text{VAE}}(\mathbf{y})}. \tag{10}$$

The evidence for minority data will be lower than for majority classes. This then motivates our inverse text densities defined as follows:

$$\psi_{\text{inv-txt-den}}(\mathbf{y}) \triangleq \frac{1}{a_3 \cdot e^{\text{ELBO}_{\text{VAE}}(\mathbf{y})}}, \tag{11}$$

where $a_3$ is a hyperparameter that controls the scale of the inverse text densities. By combining Eqn. (9) and Eqn. (11) to Eqn. (7), we can then compute $\psi$ as follows:

$$\psi = \max\left( 0, s^{a_1 + a_2 \cdot \mathbb{1}\left[\mathcal{I}(\mathbf{x}) \neq \mathcal{I}(\mathbf{x}')\right]} \right) \cdot \frac{1}{a_3 \cdot e^{\text{ELBO}_{\text{VAE}}(\mathbf{y})}} \tag{12}$$

### 3.2.4 FINAL OBJECTIVE FUNCTION

By collecting all the components discussed above, we arrive at our final training objective:

$$\mathcal{L}_{\text{final}}^{\text{PoGDiff}} = \mathbb{E}_{\mathbf{x}_0 \sim q(\mathbf{x}_0), \epsilon \sim \mathcal{N}(\mathbf{0}, \mathbf{I}), t \sim \mathcal{U}(1, T)} \left[ \|\epsilon_\theta(\mathbf{x}_t, \mathbf{y}) - \epsilon\|^2 + \psi \|\epsilon_\theta(\mathbf{x}_t, \mathbf{y}) - \epsilon_\theta(\mathbf{x}_t, \mathbf{y}')\|^2 \right], \tag{13}$$

where $\psi$ is defined in Eqn. (12). Alg. 1 summarizes our algorithm.

## 4 EXPERIMENTS

### 4.1 EXPERIMENTAL SETUP

**Datasets.** To demonstrate the effectiveness of PoGDiff in terms of both accuracy and quality, we evaluate our method on two widely used imbalanced datasets, i.e., AgeDB-IT2I (Moschoglou et al.,

Table 1: **Performance based on FID score.**

| Datasets | AgeDB-IT2I | | | | | | DigiFace-IT2I | |
|---|---|---|---|---|---|---|---|---|
| Size | Small | | Medium | | Large | | Large | |
| Metric | FID↓ | | | | | | | |
| Shot | All | Few | All | Few | All | Few | All | Few |
| VANILLA | 14.88 | 13.72 | 12.87 | 12.56 | 7.67 | 11.67 | 7.18 | 12.23 |
| CBDM | 14.72 | 14.13 | 11.63 | 11.59 | 7.18 | 11.12 | 6.96 | 12.72 |
| T2H | 14.85 | 13.66 | 12.79 | 12.52 | 7.61 | 11.64 | 7.14 | 12.22 |
| POGDIFF (OURS) | **14.15** | **12.88** | **10.89** | **10.64** | **6.03** | **10.16** | **6.84** | **11.21** |

Table 2: **Performance based on DINO score.**

| Datasets | AgeDB-IT2I | | | | | | DigiFace-IT2I | |
|---|---|---|---|---|---|---|---|---|
| Size | Small | | Medium | | Large | | Large | |
| Metric | DINO (cosine similarity) scores ↑ | | | | | | | |
| Shot | All | Few | All | Few | All | Few | All | Few |
| VANILLA | 0.42 | 0.37 | 0.39 | 0.28 | 0.34 | 0.25 | 0.42 | 0.36 |
| CBDM | 0.54 | 0.09 | 0.38 | 0.11 | 0.41 | 0.26 | 0.34 | 0.16 |
| T2H | 0.43 | 0.39 | 0.42 | 0.29 | 0.37 | 0.26 | 0.44 | 0.36 |
| POGDIFF (OURS) | **0.77** | **0.73** | **0.69** | **0.56** | **0.66** | **0.52** | **0.64** | **0.49** |

Table 3: **Performance on AgeDB-IT2I based on human evaluation.** The evaluation is a binary decision: the image is either judged as representing the same individual (score 1.0) or not (score 0.0).

| Size | Small | | Medium | | Large | |
|---|---|---|---|---|---|---|
| Metric | Human Score ↑ | | | | | |
| Shot | All | Few | All | Few | All | Few |
| VANILLA | 0.50 | 0.00 | 0.66 | 0.32 | 0.60 | 0.20 |
| CBDM | 0.50 | 0.00 | 0.44 | 0.08 | 0.56 | 0.12 |
| T2H | 0.50 | 0.00 | 0.66 | 0.32 | 0.60 | 0.20 |
| POGDIFF (OURS) | **1.00** | **1.00** | **0.96** | **0.92** | **0.84** | **0.68** |

Table 4: **Performance on AgeDB-IT2I based on GPT-4o evaluation.** The scores are from 0 to 10, with higher scores indicating the individual resembles the well-known person.

| Size | Small | | Medium | | Large | |
|---|---|---|---|---|---|---|
| Metric | GPT-4o Evaluation ↑ | | | | | |
| Shot | All | Few | All | Few | All | Few |
| VANILLA | 5.20 | 3.20 | 4.30 | 2.90 | 4.90 | 3.60 |
| CBDM | 4.50 | 1.10 | 1.30 | 1.00 | 3.10 | 1.70 |
| T2H | 5.50 | 3.10 | 4.60 | 3.00 | 4.70 | 3.90 |
| POGDIFF (OURS) | **9.10** | **8.40** | **8.80** | **8.20** | **8.50** | **8.00** |

2017) and DigiFace-IT2I (Bae et al., 2023). Note that our method is designed for fine-tuning. Therefore our setup does not require large-scale, long-tailed human datasets. Instead, we sample from these datasets, as long as they meet the following criteria: (1) the dataset must be long-tailed, (2) traditional methods must fail to recognize the minority classes, and (3) there must be a distinguishable difference between the majority and minority classes (e.g., we prefer visual distinctions between the two groups to better highlight the impact of our method). Fig. 4 shows the label density distribution of these datasets, and their level of imbalance[4].

*AgeDB-IT2I:* AgeDB-IT2I is constructed from the AgeDB dataset (Moschoglou et al., 2017). For each image $\mathbf{x}$ in AgeDB, we passed it through the pretrained LLaVA-1.6-7b model (Liu et al., 2024) to generate textual captions $\widetilde{\mathbf{y}}$. Since the identities in AgeDB are well-known individuals that the pretrained SDv1.5 (Rombach et al., 2022) might have encountered during pre-training, we masked the true names and replaced them with generic, random names, leading to a new caption $\mathbf{y}$. For example, we replace "Albert Einstein" in the caption with a random name "Lukas". Finally, we collect all $(\mathbf{y}, \mathbf{x})$ pairs to form our AgeDB-IT2I dataset.

Additionally, given that the identities (i.e., people or individuals) in AgeDB are well-known figures, we sampled from AgeDB to create three datasets for comprehensive analysis: AgeDB-IT2I-L (large), AgeDB-IT2I-M (medium), and AgeDB-IT2I-S (small). Specifically:

- *AgeDB-IT2I-L (large).* This dataset consists of 976 images across 223 identities, with each majority class containing 30 images and each minority class containing 2 images.
- *AgeDB-IT2I-M (medium).* This dataset consists of 100 images across 10 identities, with each majority class containing 30 images and each minority class containing 2 images.
- *AgeDB-IT2I-S (small).* This dataset contains 32 images across 2 identities, where each majority class consists of 30 images and each minority class consists of 2 images.

*DigiFace-IT2I-L:* DigiFace-IT2I-L is derived from the DigiFace dataset (Bae et al., 2023). It contains 985 images across 179 identities, where each majority class consists of 30 images and each minority class consists of 2 images. We use a process similar to AgeDB-IT2I to collect text-image pairs, forming this DigiFace-IT2I dataset.

**Baselines.** We employ **Stable Diffusion v1.5** (Rombach et al., 2022) as the backbone diffusion model. As this is the first work to explore imbalanced text-to-image (IT2I) diffusion models with **natural**

---

[4]Our datasets actually contain sparse datasets; more details can be found in Appendix C.4.

**SDv1.5**  **CBDM**  **PoGDiff (Ours)**

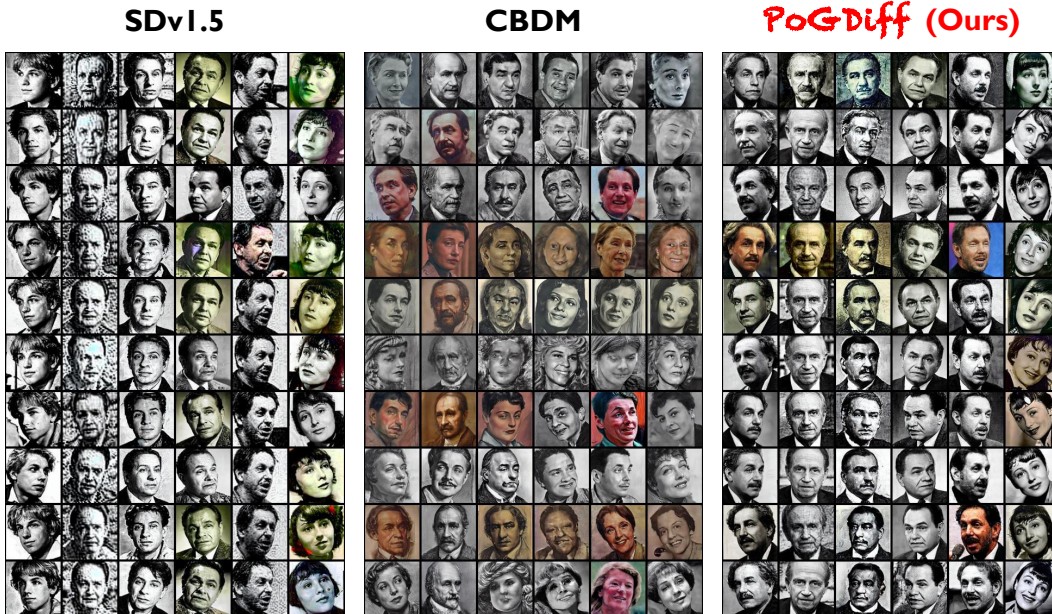

Figure 5: Example generated images from different methods. Our PoGDiff outperforms the baselines in terms of both generation accuracy and generation quality.

**text prompts**, we adapt the current state-of-the-art methods designed for long-tailed T2I diffusion models **with one-hot text prompts** to serve as baselines. The baselines are described below:

- *Vanilla:* We use term **Vanilla** to denote a model that does not incorporate any techniques for handling imbalanced data, equivalent to fine-tuning a Stable Diffusion model without additional modifications.

- *CBDM:* We use term **CBDM** to denote a model that incorporates the Class Balancing Diffusion Model (CBDM) (Qin et al., 2023) approach. During fine-tuning, we sample an additional text embedding $\mathbf{y}'$ from the entire fine-tuning dataset and apply the CBDM objective function. All hyperparameters are kept the same as in the original paper, with further details available in Qin et al. (2023).

- *T2H:* We use term **T2H** to denote a model that uses Long-Tailed Diffusion Models with Oriented Calibration (T2H) (Zhang et al., 2024). T2H is a reweighting method similar to CBDM (Qin et al., 2023), but is not directly applicable to our setting. Specifically, T2H (Zhang et al., 2024) relies on the class frequency, which is not available in our setting. In this paper, we adapted this method to our settings by using the density for each text prompt embedding to serve as the class frequency in T2H (Zhang et al., 2024).

**Evaluation Protocols and Metrics.** We use two types of evaluation metrics: generation quality and generation accuracy. For general text-to-image generation performance, we report the widely used Fréchet Inception Distance (FID) score (Heusel et al., 2017). Unlike the traditional FID score, which uses Inception-v3 (Szegedy et al., 2016) as the feature extractor, we used a pre-trained face recognition model instead; since our goal is to evaluate the ability to recognize humans, we need to capture facial features rather than general features. For each identity, we collect all images from the original AgeDB or DigiFace datasets as the *true image set*. Then, In *all-shot* evaluation, for AgeDB-IT2I-S and AgeDB-IT2I-M, we generate 100 images per identity as the *fake image set*, and for AgeDB-IT2I-L and DigiFace-IT2I-L, we generate 20 images per identity as the *fake image set*. In *few-shot* evaluation, we generate 500 images per identity as the *fake image set*. For all generations, we employ the DDIM sampling technique (Song et al., 2020a) with 50 steps. The prompt used during generation is "An image of {p}." where "p" is the name of the identity (e.g., Albert Einstein).

To assess generation accuracy, we use 10 different seeds to sample 10 images for each minority class. We then gather feedback from both the GPT-4o model (Achiam et al., 2023) and human evaluators to score the accuracy of identity recognition. Additionally, we employ a pre-trained DINO model (Caron et al., 2021) for calculating DINO score for image similarities. More details about the evaluation process including prompts we used are in the Appendix.

**Implementation Details.** For both baselines and our model, we used the same hyper-parameter settings, specifically

- *AgeDB-IT2I-L* & *DigiFace-IT2I-L*. The learning rate was set to $1 \times 10^{-5}$, with a maximum of $12,000$ training steps. The effective batch size per GPU was 32, calculated as 8 (Batch Size) $\times$ 4 (Gradient Accumulation Steps).
- *AgeDB-IT2I-M* & *AgeDB-IT2I-S*. The learning rate was set to $1 \times 10^{-5}$, with a maximum of $6,000$ training steps. The effective batch size per GPU was 8, calculated as 8 (Batch Size) $\times$ 1 (Gradient Accumulation Step).

## 4.2 RESULTS

We report the performance of different methods in terms of FID score, human evaluation score, GPT-4o score, and DINO score in Table 1, Table 2, Table 3 and Table 4, respectively[5]. Across all tables, we observe that our PoGDiff consistently outperforms all baselines. Notably, PoGDiff demonstrates significant improvements, especially in few-shot scenarios (i.e., for minority classes). It is also worth noting that CBDM (Qin et al., 2023) performs extremely poorly on AgeDB-IT2I-S and AgeDB-IT2I-M datasets. This is because their method samples text conditions from the entire space, which may work in one-hot class settings, but in our context (natural text conditions), this sampling technique misguides the model during training. In addition, for each method, we report the performance on low-density classes in AgeDB-IT2I-L in Fig. 5. Across each column, the individual names are Albert Einstein, JW Marriott, J.P. Morgan, Edward G. Robinson, Larry Ellison, and Luise Rainer, respectively. The results show that our PoGDiff achieves significantly better accuracy and quality for tail classes.

Note that one of our primary objectives is to generate accurate images of the same individual while ensuring facial consistency. Therefore **diversity can sometimes be harmful**. For example, given a text input of "Einstein", generated images with high diversity would generate both male and females images; **this is obviously incorrect**. Therefore it is important to strike a balance between **diversity** and **accuracy**, a goal that our PoGDiff achieves.

Specifically, as shown in Fig. 5:

- **First Three Columns of SDv1.5, CBDM, and PoGDiff**: In these cases, the **training** dataset contains **only two images per person**[6]. With such limited data, it is impossible to introduce meaningful diversity.
  - SDv1.5 fails to generate accurate images altogether in this scenario.
  - While CBDM might appear to produce the "diversity" you mentioned, it does so incorrectly, as it generates an image of a woman when the target is Einstein.
  - In contrast, our PoGDiff can successfully generate accurate images (e.g., Einstein images in Column 1) while still enjoying sufficient diversity.
- **Fourth and Fifth Columns**: Here, the training dataset contains a medium number of images per person (5–7 images). Under these conditions:
  - SDv1.5 can generate accurate representations of individuals, but its outputs lack diversity.
  - CBDM, on the other hand, introduces "diversity" but consistently generates incorrect results.
  - In contrast, our method produces accurate images of the target individual while demonstrating greater diversity than SDv1.5.
- **Sixth Column**: In this case, the training dataset includes 30 images per person.
  - SDv1.5 generates accurate images but with nearly identical expressions, i.e., poor diversity.
  - CBDM still fails to generate accurate depictions of the individual.
  - In contrast, PoGDiff successfully generates accurate images while maintaining diversity.

In summary, typical diversity evaluation in diffusion model evaluations, such as generating multiple types of trees for a "tree" prompt, is **not the focus of our setting** and may even be **misleading**. In our setting, the key is to balance accuracy and diversity.

---

[5]Note that the CLIP score is not applicable in our setting. Specifically, our text prompts are predominantly human names. However, CLIP is primarily trained on common objects, not human names; therefore the CLIP score can not be use to compute matching scores between images and human names.

[6]More details are included in the Appendix.

## 5 ABLATION STUDY

To verify the effectiveness of each component in the second term in our PoGDiff final objective function from Eqn. (12), we report the accuracy of our proposed PoGDiff after removing the $\mathbf{y}'$ (i.e., same as Vanilla model), the Image Similarity term $\psi_{\text{img-sim}}$, and/or the Inverse Text Densities term $\psi_{\text{inv-txt-den}}$ in Table 5 for AgeDB-IT2I-L. The results show that removing either term leads to a performance drop, confirming the importance of both terms in our PoGDiff.

## 6 LIMITATIONS

**Datasets.** Our method relies heavily on "borrowing" the statistical strength of neighboring samples from minority classes, making the results sensitive to the size of the minority class. (i.e., in our assumption we require **at least** 2 for each minority class). In addition, while our AgeDB-IT2I-small and AgeDB-IT2I-medium are actually the sparse dataset, the cardinality remains limited in our experiments. Therefore, how to address IT2I problem under this settings are interesting directions.

Table 5: **Ablation Studies.**

| Datasets | AgeDB-IT2I-Large | | | | | | | |
|---|---|---|---|---|---|---|---|---|
| Size | FID $\downarrow$ | | Human $\uparrow$ | | GPT-4o $\uparrow$ | | DINO $\uparrow$ | |
| Shot | All | Few | All | Few | All | Few | All | Few |
| W/O $\mathbf{y}'$ (VANILLA) | 7.67 | 11.67 | 0.60 | 0.20 | 4.90 | 3.60 | 0.34 | 0.25 |
| W/O $\psi_{\text{IMG-SIM}}$ | 6.41 | 10.49 | **0.84** | **0.68** | 8.40 | 7.60 | 0.57 | 0.46 |
| W/O $\psi_{\text{INV-TXT-DEN}}$ | 6.35 | 10.43 | **0.84** | **0.68** | 8.20 | 7.80 | 0.64 | 0.51 |
| POGDIFF (OURS) | **6.03** | **10.16** | **0.84** | **0.68** | **8.50** | **8.00** | **0.66** | **0.52** |

**Models.** Our method is a general fine-tuning approach designed for datasets that the Stable Diffusion (SD) model has not encountered during pre-training. As shown in Fig. 1, color deviation is very common and is a known issue when one fine-tunes diffusion models (as also mentioned in (Song et al., 2020b)); for example, we can observe similar color deviation in both baselines (e.g., CBDM and Stable Diffusion v1.5) and our PoGDiff. This can be mitigated using the exponential moving average (EMA) technique (Song et al., 2020b); however, this is orthogonal to our method and is outside the scope of our paper. Moreover, as shown in Fig. 5, the baseline Stable Diffusion also suffers from this issue. Besides, exploring PoGDiff's performance when training from scratch is also an interesting direction for future work.

**Methodology.** The distance between the current text embedding $\mathbf{y}$ and the sampled $\mathbf{y}'$ impacts the final generated results, therefore in our paper, we introduced a more sophisticated approach for computing the weight $\psi$, which depends on the quality of the image pre-trained model and our trained VAE. These mechanisms ensure that data points with smaller distances are assigned higher effective weights. Effectively producing $\psi$ for any new, arbitrary dataset remains an open question and is an interesting avenue for future work, as it could further enhance the method's performance.

**Evaluation.** Our goal is to adapt the pretrained diffusion model to a specific dataset; therefore the evaluation should focus on the target dataset rather than the original dataset used during pre-training. For example, when a user fine-tunes a model on a dataset of employee faces, s/he is not interested in how well the fine-tuned model can generate images of "tables" and "chairs". Evaluating the model's performance on the original dataset used during pre-training would be an intriguing direction for future work, but it is orthogonal to our proposed PoGDiff and out of the scope of our paper.

## 7 CONCLUSIONS

In this paper, we propose a general fine-tuning approach called PoGDiff to address the performance drop that occurs when fine-tuning on imbalanced datasets. Instead of directly minimizing the KL divergence between the predicted and ground-truth distributions, PoGDiff replaces the ground-truth distribution with a Product of Gaussians (PoG), constructed by combining the original ground-truth targets with the predicted distribution conditioned on a neighboring text embedding. Looking ahead, an interesting avenue for future research would be to explore more innovative techniques for re-weighting minority classes (as discussed in Sec. 6), particularly within the constraints of: (1) long-tailed generation settings, as opposed to recognition tasks, and (2) natural text prompts rather than one-hot class labels.

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

## A PROOFS FOR PROPOSITION 3.1

**Proposition A.1.** *Assume $\lambda_{\mathbf{y}} = \lambda_{PoG} \triangleq \lambda_t + \lambda_{\mathbf{y}'}$, we have our loss function*

$$\mathcal{L}_{t-1}^{PoGDiff} = \mathbb{E}_q \left[ \frac{\lambda_{\mathbf{y}}}{2} \|\mu_\theta(\mathbf{x}_t, \mathbf{y}) - \mu_{PoG}\|^2 \right] + C. \tag{14}$$

*Here, $C$ is a constant, and $\mu_{PoG}$ denotes the mean of the PoG, with the expression defined in Eqn. (2). Then, through derivations based on Gaussian properties, we obtain*

$$\mathcal{L}_{t-1}^{PoGDiff} \leq \mathbb{E}_q \left[ \mathcal{A}(\lambda_t) \|\epsilon_\theta(\mathbf{x}_t, \mathbf{y}) - \epsilon\|^2 + \mathcal{A}(\lambda_{\mathbf{y}'}) \|\epsilon_\theta(\mathbf{x}_t, \mathbf{y}) - \epsilon_\theta(\mathbf{x}_t, \mathbf{y}')\|^2 \right] + C \tag{15}$$

*where the function $\mathcal{A}(\lambda) \triangleq \dfrac{\lambda(1-\alpha_t)^2}{2\alpha_t(1-\bar{\alpha}_t)}$.*

*Proof.* To prove the above inequality, we need to prove the following lemma.

**Lemma A.1.** *Assume $\lambda_{\mathbf{y}} = \lambda_{PoG} \triangleq \lambda_t + \lambda_{\mathbf{y}'}$, and for simplicity we shorten the notation from $\epsilon_\theta(\mathbf{x}_t, \mathbf{y})$ and $\mu_\theta(\mathbf{x}_t, \mathbf{y})$ to $\epsilon_\theta(\mathbf{y})$ and $\mu_\theta(\mathbf{y})$, respectively. Then we have*

$$\frac{1}{2}\lambda_t \left(\mu_\theta(\mathbf{y}) - \mu_t\right)^2 + \frac{1}{2}\lambda_{\mathbf{y}'} \left(\mu_\theta(\mathbf{y}) - \mu_\theta(\mathbf{y}')\right)^2 \geq \frac{1}{2}\lambda_{\mathbf{y}} \left(\mu_\theta(\mathbf{y}) - \mu_{PoG}\right)^2 \tag{16}$$

*Proof.* By the definition of Gaussian property, we have

$$\frac{1}{2}\lambda_t \left(\mu_\theta(\mathbf{y}) - \mu_t\right)^2 + \frac{1}{2}\lambda_{\mathbf{y}'} \left(\mu_\theta(\mathbf{y}) - \mu_\theta(\mathbf{y}')\right)^2$$

$$= \frac{[\mu_\theta(\mathbf{y})]^2 - 2\mu_t\mu_\theta(\mathbf{y}) + \mu_t^2}{2\lambda_t^{-1}} + \frac{[\mu_\theta(\mathbf{y})]^2 - 2\mu_\theta(\mathbf{y}')\mu_\theta(\mathbf{y}) + [\mu_\theta(\mathbf{y}')]^2}{2\lambda_{\mathbf{y}'}^{-1}}$$

$$= \frac{\left(\lambda_t^{-1} + \lambda_{\mathbf{y}'}^{-1}\right)[\mu_\theta(\mathbf{y})]^2 - 2\left(\dfrac{\mu_t}{\lambda_{\mathbf{y}'}} + \dfrac{\mu_\theta(\mathbf{y}')}{\lambda_t}\right)\mu_\theta(\mathbf{y}) + \dfrac{\mu_t^2}{\lambda_{\mathbf{y}'}} + \dfrac{[\mu_\theta(\mathbf{y}')]^2}{\lambda_t}}{2[\lambda_t\lambda_{\mathbf{y}'}]^{-1}}$$

$$= \frac{[\mu_\theta(\mathbf{y})]^2 - 2\left(\dfrac{\mu_t\lambda_t + [\mu_\theta(\mathbf{y}')]\lambda_{\mathbf{y}'}}{\lambda_t + \lambda_{\mathbf{y}'}}\right)\mu_\theta(\mathbf{y}) + \dfrac{\mu_t^2\lambda_t + [\mu_\theta(\mathbf{y}')]^2\lambda_{\mathbf{y}'}}{\lambda_t + \lambda_{\mathbf{y}'}}}{\dfrac{2}{\lambda_t + \lambda_{\mathbf{y}'}}}$$

$$+ \frac{\left[\dfrac{\mu_t\lambda_t + [\mu_\theta(\mathbf{y}')]\lambda_{\mathbf{y}'}}{\lambda_t + \lambda_{\mathbf{y}'}}\right]^2}{\dfrac{2}{\lambda_t + \lambda_{\mathbf{y}'}}} - \frac{\left[\dfrac{\mu_t\lambda_t + [\mu_\theta(\mathbf{y}')]\lambda_{\mathbf{y}'}}{\lambda_t + \lambda_{\mathbf{y}'}}\right]^2}{\dfrac{2}{\lambda_t + \lambda_{\mathbf{y}'}}}$$

$$= \frac{\left(\mu_\theta(\mathbf{y}) - \dfrac{\mu_t\lambda_t + [\mu_\theta(\mathbf{y}')]\lambda_{\mathbf{y}'}}{\lambda_t + \lambda_{\mathbf{y}'}}\right)^2}{\dfrac{2}{\lambda_t + \lambda_{\mathbf{y}'}}} + \frac{(\mu_t^2\lambda_t + [\mu_\theta(\mathbf{y}')]^2\lambda_{\mathbf{y}'})(\lambda_t + \lambda_{\mathbf{y}'}) - (\mu_t\lambda_t + [\mu_\theta(\mathbf{y}')]\lambda_{\mathbf{y}'})^2}{2(\lambda_t + \lambda_{\mathbf{y}'})}$$

$$= \frac{1}{2}\lambda_{\mathbf{y}} \left(\mu_\theta(\mathbf{y}) - \mu_{\text{PoG}}\right)^2 + \frac{\lambda_t\lambda_{\mathbf{y}'}(\mu_t - \mu_\theta(\mathbf{y}'))^2}{2(\lambda_t + \lambda_{\mathbf{y}'})}$$

$$\geq \frac{1}{2}\lambda_{\mathbf{y}} \left(\mu_\theta(\mathbf{y}) - \mu_{\text{PoG}}\right)^2.$$

Thus we complete the proof. $\square$

From Lemma A.1, we can derive

$$
\begin{aligned}
\frac{1}{2}\lambda_{\mathbf{y}}\left\|\mu_\theta(\mathbf{y}) - \mu_{\mathrm{PoG}}\right\|^2 &\equiv \frac{1}{2}\lambda_{\mathbf{y}}\left(\mu_\theta(\mathbf{y}) - \mu_{\mathrm{PoG}}\right)^2 \\
&\leq \frac{1}{2}\lambda_t\left(\mu_\theta(\mathbf{y}) - \mu_t\right)^2 + \frac{1}{2}\lambda_{\mathbf{y}'}\left(\mu_\theta(\mathbf{y}) - \mu_\theta(\mathbf{y}')\right)^2 \\
&\equiv \frac{1}{2}\lambda_t\left\|\mu_\theta(\mathbf{y}) - \mu_t\right\|^2 + \frac{1}{2}\lambda_{\mathbf{y}'}\left\|\mu_\theta(\mathbf{y}) - \mu_\theta(\mathbf{y}')\right\|^2 \\
&\equiv \mathcal{A}(\lambda_t)\left\|\epsilon_\theta(\mathbf{y}) - \epsilon\right\|^2 + \mathcal{A}(\lambda_{\mathbf{y}'})\left\|\epsilon_\theta(\mathbf{y}) - \epsilon_\theta(\mathbf{y}')\right\|^2,
\end{aligned}
$$

where the function $\mathcal{A}(\lambda) \triangleq \dfrac{\lambda(1 - \alpha_t)^2}{2\alpha_t(1 - \bar{\alpha}_t)}$, and the last equivalence is because the transform from $\mu_\theta(\cdot)$ to $\epsilon_\theta(\cdot)$. □

## B  DETAILS FOR EVALUATION

In this section, we provide details on our evaluation procedures.

**FID Score.** For each identity, we collect all images from the original AgeDB or DigiFace datasets as the *true image set*. Then, In *all-shot* evaluation, for AgeDB-IT2I-S and AgeDB-IT2I-M, we generate 100 images per identity as the *fake image set*, and for AgeDB-IT2I-L and DigiFace-IT2I-L, we generate 20 images per identity as the *fake image set*. In *few-shot* evaluation, we we generate 500 images per identity as the *fake image set*. For all generations, we employ the DDIM sampling technique (Song et al., 2020a) with 50 steps. The prompt used during generation is "An image of { p }." where "p" is the name of the identity (e.g., Albert Einstein).

**Human & GPT-4o Feedback.** For each minority identity, we generate 5 images using DDIM sampling (Song et al., 2020a) with 50 steps. We then ask 10 people to evaluate whether the images depict the same person (scored as $1.0$) or not (scored as $0.0$). Additionally, for each image, we ask the GPT-4o model to rate the similarity on a scale from 1 to 10. The prompt used during generation is "An image of { p }." where "p" is the name of the identity. The text prompt using for GPT-4o model is "It is mandatory to give a score that how close the person in the image to a well-known individual. A score of 10.0 means they are exactly the same person, while a score of 0.0 means they are definitely not the same person. How close you think the person in the image is to 'p-true'." where "p-true" denotes the real name (well-known name) in AgeDB. Note that the GPT-4o model might occasionally refuse to provide a score, and you may need to repeat and compel it to give a rating. For each image, we collect 10 scores from the GPT-4o model and report the average rating.

**Evaluating Image Similarities.** We collect samples that are outside our training dataset (e.g., AgeDB-T2I-L) but belong to the original dataset (e.g., AgeDB). Using the same prompt, we generate the corresponding images. A pre-trained DINOv2 model (Caron et al., 2021) is then applied to extract latent features, and cosine similarities are calculated.

## C  DISCUSSION

### C.1  PROBLEM SETTINGS

We would like to clarify that our paper focuses on a setting different from works like Dream-Booth (Ruiz et al., 2023), and our focus is not on diversity, but on finetuning a diffusion model on an imbalanced dataset. Specifically:

- **Different Setting from Custom Techniques like DreamBooth (Ruiz et al., 2023), CustomDiffusion (Kumari et al., 2023) and PhotoMaker (Li et al., 2024).** Previous works like CustomDiffusion and PhotoMaker focus on adjusting the model to generate images with **a single object**, e.g., a specific dog. In contrast, our PoGDiff focuses finetuning the diffusion model on an entire data with **many different objects/persons simultaneously**. They are **very different settings** and are **complementary** to each other.

- **Diversity.** Note that while our PoG can naturally generate images with diversity, diversity is actually **not** our focus. Our goal is to fine-tune a diffusion model on an imbalanced dataset.

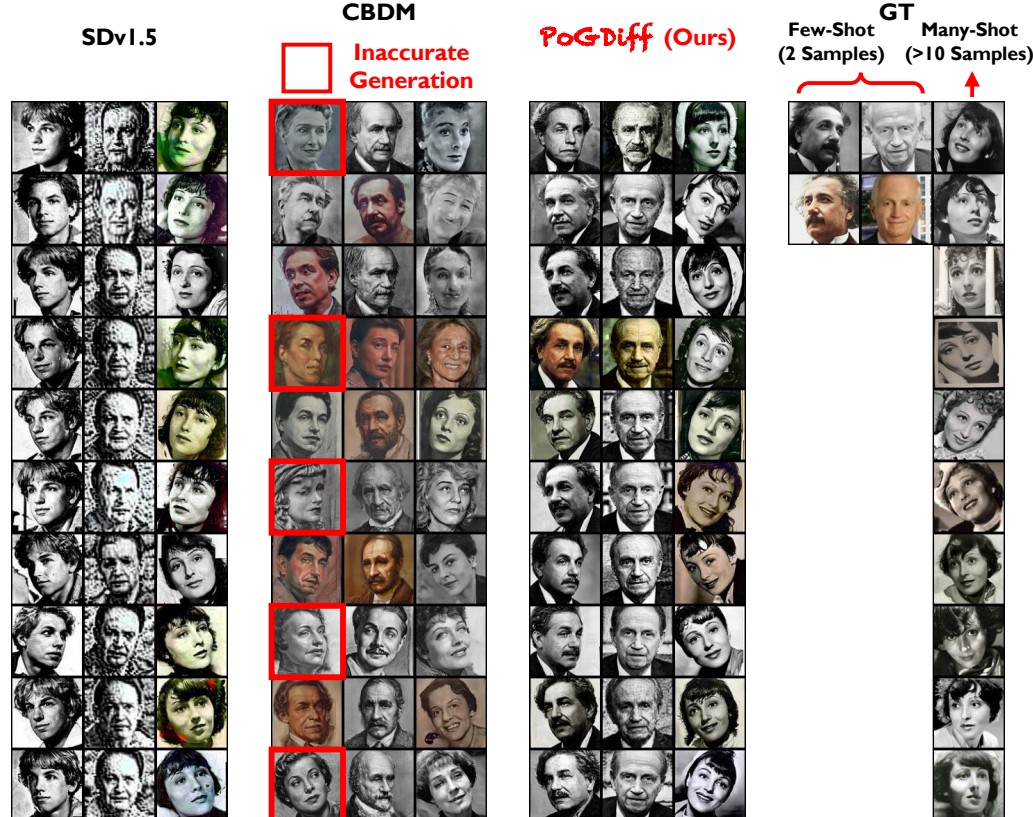

Figure 6: Example generated images from different methods. Our PoGDiff outperforms the baselines in both generation accuracy and quality. Regarding the ground truth (GT), the training set for the minority class (left two columns) contains only 2 images per individual, whereas the majority class has more than 10 samples per individual.

> For example, PoGDiff can fine-tune a diffusion model on an imbalanced dataset of employee faces so that the diffusion model can generate new images that match each employee's identity. In this case, we are more interested in "faithfulness" rather than "diversity".

## C.2 UNDERSTANDING FIG. 5

To further emphasize our results, note that one of our primary objectives is to generate accurate images of the same individual while ensuring facial consistency. Therefore **diversity can be harmful**. For example, given a text input of "Einstein", generated images with high diversity would generate both male and females images; **this is obviously incorrect**. Therefore it is important to strike a balance between **diversity** and **accuracy**, a goal that our PoGDiff achieves.

Fig. 6 (which contains images from Column 1, 2, and 6 for each method in Fig. 5) provides a clearer comparison with the training images. Specifically:

- **Ground-Truth (GT) Images**: We show the ground-truth images on the right-most 3 columns.

- **Column 1 and 2 of SDv1.5, CBDM, PoGDiff, and GT**: In these cases, the **training** dataset contains **only two images per person**. With such limited data, it is impossible to introduce meaningful diversity.
    - SDv1.5 fails to generate accurate images altogether in this scenario.
    - While CBDM might appear to produce the "diversity" you mentioned, it does so incorrectly, as it generates an image of a woman when the target is Einstein (we circled those wrong samples in first column in Fig. 6).

| SDv1.5 | CBDM | T2H | PoGDiff (Ours) |
|---|---|---|---|

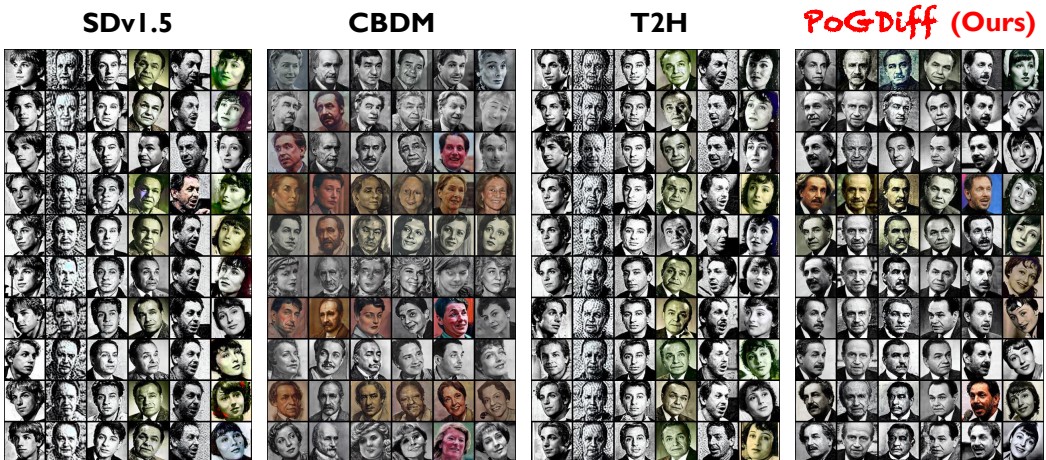

Figure 7: Example generated images from different methods. Our PoGDiff outperforms the baselines in terms of both generation accuracy and generation quality.

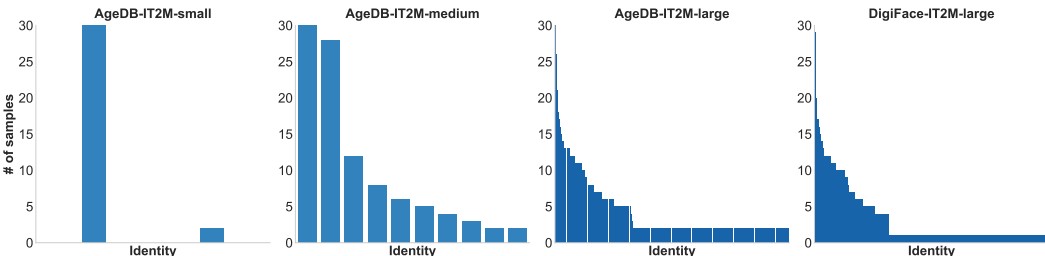

Figure 8: Overview of label distribution for four IT2I datasets in bar plots. The x-axis corresponds to the identities (i.e., people or individuals).

- In contrast, our PoGDiff can successfully generate accurate images (e.g., Einstein images in Column 1) while still enjoying sufficient diversity.
- **Column 3 of of SDv1.5, CBDM, PoGDiff, and GT**: In this case, the training dataset includes around 30 images per person.
  - SDv1.5 generates accurate images but with nearly identical expressions, offering minimal diversity.
  - CBDM still fails to generate accurate depictions of the individual.
  - In contrast, our PoGDiff successfully generates accurate images while introducing notable diversity.

In summary, typical diversity evaluation in diffusion model evaluations, such as generating multiple types of trees for a "tree" prompt, is **not the focus of our setting** and may even be **misleading**. In our setting, the key is to balance accuracy and diversity.

## C.3 WHY NOT DIRECTLY SMOOTH TEXT EMBEDDING?

Preliminary results indicate that directly smoothing the text embeddings does not yield meaningful improvements. Below we provide some insights into why this approach might fail. Suppose we have a text embedding $y$ and its corresponding neighboring embedding $y'$. Depending on their relationship, we are likely to encounter three cases:

- **Case 1: $\mathbf{y}' = \mathbf{y}$.** In this case, applying a reweighting method such as a linear combination results in no meaningful change, as the smoothing outcome is still $\mathbf{y}$.
- **Case 2: $\mathbf{y}'$ is far from $\mathbf{y}$.** If $\mathbf{y}'$ is significantly distant from $\mathbf{y}$, combining them becomes irrelevant and nonsensical, as $\mathbf{y}'$ no longer represents useful neighboring information.
- **Case 3: $\mathbf{y}'$ is very close to $\mathbf{y}$.** When $\mathbf{y}'$ is close to $\mathbf{y}$, the reweighting can be approximated as: $\alpha \mathbf{y} + (1 - \alpha)\mathbf{y}' \approx \mathbf{y} + (1 - \alpha)(\mathbf{y}' - \mathbf{y})$. Since $\mathbf{y}'$ is nearly identical to $\mathbf{y}$, this

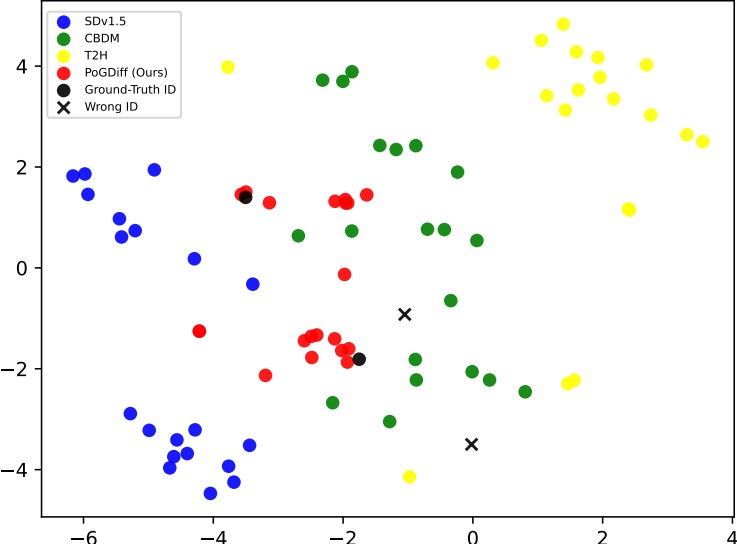

Figure 9: TSNE visualization for all the methods for an example individual in the Age-DB-IT2M-large dataset.

effectively introduces a small weighted noise term $(1-\alpha)(\mathbf{y}'-\mathbf{y})$ into $\mathbf{y}$. In our preliminary experiments, this additional noise degraded the performance compared to the original baseline results.

Based on these observations, direct smoothing of text embeddings appears ineffective and may even harm performance in some cases.

## C.4 OUR DATASET COVERS DIFFERENT LEVELS OF SPARSITY

Our AgeDB-IT2M-small and AgeDB-IT2M-medium datasets are actually very sparse and are meant for evaluate the sparse data. For example, the AgeDB-IT2M-small only contains images from 2 persons, it is therefore a very sparse data setting, compared to AgeDB-IT2M-large with images across 223 persons. Fig. 8 shows the bar plot version for our datasets, while sparse settings are not our primary focus, we agree that addressing imbalanced image generation in such setting is an interesting and valuable direction, and we have included a discussion about this in the limitations section of the paper.

## C.5 DISCUSSION ON FID

It is important to note that the FID score measures only the distance between Gaussian distributions of ground-truth and generated images, relying solely on mean and variance. As a result, it does not fully capture the nuances of our task. This is why we include additional evaluation metrics such as DINO Score, Human Score, and GPT-4o Score, to comprehensively verify our method's superiority (as shown in Table 2, Table 3 and Table 4).

**Additional Experiments: Limitation of FID.** In addition, we have added a figure showcasing a t-SNE visualization for a minority class as an example, as shown in Fig. 9, to further illustrate the limitation of FID we mentioned above. As shown in the figure:

- There are two ground-truth IDs (i.e., two ground-truth individuals) in the training set.

- Our PoGDiff can successfully generate images similar to these two ground-truth ID while maintaining diversity.

- All baselines, including CBDM, fail to generate accurate images according to the ground-truth IDs. In fact most generated images from the baselines are similar to other IDs, i.e., generating the facial images of wrong individuals.

These results show that:

- Our PoGDiff significantly outperforms the baselines.
- FID fails to capture such improvements because it depends only on the mean and variance of the distribution, losing a lot of information during evaluation.

## D    MORE BASELINES

In this section, we also include one more work related to our baseline CBDM (Qin et al., 2023); they are both equivalent to direct reweighting/resampling. We did not include (Zhang et al., 2024) as a baseline because it is not directly applicable to our setting. Specifically, (Zhang et al., 2024) relies on the class frequency, which is not available in our setting. Therefore, we adapted this method to our settings by using the density for each text prompt embedding to serve as the class frequency in (Zhang et al., 2024). Results shown in Fig. 7 show that it performs even worse than CBDM, and it performs similar to directly fine-tuning a SD model.

## E    MORE EVALUATION METRIC: RECALL SCORE.

**FID Measures Both ID Consistency and Diversity.** We could like to clarify that our Fréchet Inception Distance (FID) is computed for each ID separately, and the final FID score in the tables (e.g., Table 1) is the average FID over all IDs. Therefore FID measures both ID consistency and diversity.

To see why, note that the FID score measures the distance between two Gaussian distributions, where the *mean* of the Gaussian represents the *identity (ID)* and the *variance* represents the *diversity*. For example, the

Table 6: **Recall for the AgeDB-IT2I Dataset.** See the detailed definition of recall in Appendix E.

| Size | Small | | Medium | | Large | |
|---|---|---|---|---|---|---|
| Metric | Recall Score ↑ | | | | | |
| Shot | All | Few | All | Few | All | Few |
| VANILLA | 0.017 | 0.000 | 0.104 | 0.167 | 0.196 | 0.200 |
| CBDM | 0.267 | 0.000 | 0.159 | 0.083 | 0.138 | 0.100 |
| T2H | 0.017 | 0.000 | 0.104 | 0.167 | 0.196 | 0.200 |
| POGDIFF (OURS) | **0.800** | **1.000** | **0.517** | **0.642** | **0.435** | **0.540** |

*mean* of the ground-truth distribution represents the embedding position of the ground-truth ID, while the *variance* of the ground-truth distribution represents the *diversity* of ground-truth images. Similarly, the *mean* of the generated-image distribution represents the embedding position of the generated-image ID, while the *variance* of the generated-image distribution represents the *diversity* of generated images. A lower FID score indicates that the generated-image distribution more closely matches the ground truth distribution **in terms of both ID and diversity**.

**Results Related to Diversity.** Currently,

- **PoGDiff's Superior FID Performance.** as shown in Table 1, we demonstrate that PoGDiff achieves a lower FID score, particularly in few-shot regions (i.e., minorities). This suggests that the images generated by our method capture a broader range of variations present in the training dataset, such as **backgrounds or facial angles**.
- **PoGDiff's Visualization.** As shown in Fig. 6:
  - For Einstein (Column 1 for each method), the training dataset includes two face angles and two hairstyles. Our generated results successfully cover these attributes.
  - For JW Marriott (Column 2 for each method), the training dataset has only one face angle. Correspondingly our results focus on generating subtle variations in facial expressions with only one angle, **as expected**.
  - For the majority group (Column 3 for each method), our results clearly show that the generated images cover a wider range of diversity while maintaining ID consistency.

**Additional Experiments on Recall (a New Metric).**   To better evaluate the superiority of our PoGDiff, we propose a new metric, "recall".

- **Recall in the Context of Image Generation: "Correct Image" and "Covered Image".** For each generated image, we classify it as a "correct image" if its distance to at least one

ground-truth (GT) image is below a predefined threshold. For instance, suppose we have two training-set images for Einstein, denoted as $x_1$ and $x_2$. A generated image $x_g$ is a "correct image" if the cosine similarity between $x_g$ and either $x_1$ or $x_2$ is above some threshold (e.g., we set to 0.9 here). For example, if the cosine similarity $x_g$ and $x_1$ is larger than 0.9, we say that $x_g$ is a "correct image", and that $x_1$ is a "covered image". Intuitively, a training-set image (e.g., $x_1$) is covered if a diffusion model is capable of generating a similar image.

- **Formal Definition for Recall.** Formally, for each model, we compute the **Recall** per ID as follows:

$$\text{Recall} = \frac{1}{c} \sum_{i=1}^{c} \frac{\text{number of unique covered images for ID i}}{\text{number of images for ID i in the training set}} \tag{17}$$

where $c$ is the number of IDs in a training set.

- **Cosine Similarity between Images.** Note that in practice, we compute the cosine similarity between DINO embeddings of images rather than raw pixels.

- **Analysis.** This metric evaluates the generational diversity of a model. For example, if the training dataset contains two distinct images of Einstein, $x_1$ and $x_2$, and a model generates only images resembling $x_1$, the recall in this case would be 0.5. While the model may achieve high accuracy in terms of facial identity ( Table 3 and  Table 4), it falls short in diversity because it fails to generate images resembling $x_2$. In contrast, if a model generates images that cover both $x_1$ and $x_2$ the recall for this ID will be 1; for instance, if the model generates 10 images for Einstein, where 6 of them resemble $x_1$ and 4 of them resemble $x_2$, the recall would be 1, indicating high diversity and coverage.

**Additional Results in Terms of Recall.** Table 6 shows the recall for different methods on three datasets, AgeDB-IT2I-small, AgeDB-IT2I-medium, and AgeDB-IT2I-large. These results show that our PoGDiff achieves much higher recall compared to all baselines, demonstrating its impressive diversity.

**Additional Details for AgeDB-IT2I-small in Table 6.** For AgeDB-IT2I-small, there are two IDs, one "majority" ID with 30 images and one minority ID with 2 images.

- For **VANILLA** and **T2H**, the recall for the majority ID and the minority ID is $1/30$ and $0/2$, respectively. Therefore, the average recall score is $0.5 * 1/30 + 0.5 * 0/2 \approx 0.0167$.

- For **CBDM**, the recall for the majority ID and the minority ID is $16/30$ and $0/2$, respectively. Therefore, the average recall score is $0.5 * 16/30 + 0.5 * 0/2 \approx 0.2667$.

- For **PoGDiff (Ours)**, the recall for the majority ID and the minority ID is $18/30$ and $2/2$, respectively. Therefore, the average recall score is $0.5 * 18/30 + 0.5 * 2/2 = 0.8$.

## F    MORE DATASETS

We have included an additional dataset, VGGFace, for evaluation. Specifically, we constructed a subset from VGGFace2 (Cao et al., 2018), named **VGGFace-IT2I-small**. This is a **sparse** dataset consisting of two individuals: the majority group contains 30 images, while the minority group contains only 2 images.

The results shown in Table 7,  Table 8,  Table 9,  Table 10 and  Table 11, below demonstrate that our **PoGDiff** consistently outperform all baselines, highlighting its robustness and superior performance even on **imbalanced and sparse** datasets.

Table 7: **Performance based on FID score.**

| Datasets | AgeDB-IT2I | | | | | | DigiFace-IT2I | | VGGFace-IT2I | |
|---|---|---|---|---|---|---|---|---|---|---|
| Size | Small | | Medium | | Large | | Large | | Small | |
| Metric | FID ↓ | | | | | | | | | |
| Shot | All | Few | All | Few | All | Few | All | Few | All | Few |
| VANILLA | 14.88 | 13.72 | 12.87 | 12.56 | 7.67 | 11.67 | 7.18 | 12.23 | 14.18 | 12.73 |
| CBDM | 14.72 | 14.13 | 11.63 | 11.59 | 7.18 | 11.12 | 6.96 | 12.72 | 13.85 | 13.21 |
| T2H | 14.85 | 13.66 | 12.79 | 12.52 | 7.61 | 11.64 | 7.14 | 12.22 | 14.16 | 12.74 |
| POGDIFF (OURS) | **14.15** | **12.88** | **10.89** | **10.64** | **6.03** | **10.16** | **6.84** | **11.21** | **13.68** | **11.11** |

Table 8: **Performance based on DINO score.**

| Datasets | AgeDB-IT2I | | | | | | DigiFace-IT2I | | VGGFace-IT2I | |
|---|---|---|---|---|---|---|---|---|---|---|
| Size | Small | | Medium | | Large | | Large | | Small | |
| Metric | DINO (cosine similarity) scores ↑ | | | | | | | | | |
| Shot | All | Few | All | Few | All | Few | All | Few | All | Few |
| VANILLA | 0.42 | 0.37 | 0.39 | 0.28 | 0.34 | 0.25 | 0.42 | 0.36 | 0.49 | 0.36 |
| CBDM | 0.54 | 0.09 | 0.38 | 0.11 | 0.41 | 0.26 | 0.34 | 0.16 | 0.52 | 0.06 |
| T2H | 0.43 | 0.39 | 0.42 | 0.29 | 0.37 | 0.26 | 0.44 | 0.36 | 0.48 | 0.37 |
| POGDIFF (OURS) | **0.77** | **0.73** | **0.69** | **0.56** | **0.66** | **0.52** | **0.64** | **0.49** | **0.84** | **0.79** |

Table 9: **Performance on AgeDB-IT2I based on human evaluation.** The evaluation is a binary decision: the image is either judged as representing the same individual (score 1.0) or not (score 0.0).

| Datasets | AgeDB-IT2I | | | | | | VGGFace-IT2I | |
|---|---|---|---|---|---|---|---|---|
| Size | Small | | Medium | | Large | | Small | |
| Metric | Human Score ↑ | | | | | | | |
| Shot | All | Few | All | Few | All | Few | All | Few |
| VANILLA | 0.50 | 0.00 | 0.66 | 0.32 | 0.60 | 0.20 | 0.50 | 0.00 |
| CBDM | 0.50 | 0.00 | 0.44 | 0.08 | 0.56 | 0.12 | 0.50 | 0.00 |
| T2H | 0.50 | 0.00 | 0.66 | 0.32 | 0.60 | 0.20 | 0.50 | 0.00 |
| POGDIFF (OURS) | **1.00** | **1.00** | **0.96** | **0.92** | **0.84** | **0.68** | **1.00** | **1.00** |

Table 10: **Performance on AgeDB-IT2I based on GPT-4o evaluation.** The scores are from 0 to 10, with higher scores indicating the individual resembles the well-known person.

| Datasets | AgeDB-IT2I | | | | | | VGGFace-IT2I | |
|---|---|---|---|---|---|---|---|---|
| Size | Small | | Medium | | Large | | Small | |
| Metric | GPT-4o Evaluation ↑ | | | | | | | |
| Shot | All | Few | All | Few | All | Few | All | Few |
| VANILLA | 5.20 | 3.20 | 4.30 | 2.90 | 4.90 | 3.60 | 6.00 | 3.60 |
| CBDM | 4.50 | 1.10 | 1.30 | 1.00 | 3.10 | 1.70 | 4.67 | 1.33 |
| T2H | 5.50 | 3.10 | 4.60 | 3.00 | 4.70 | 3.90 | 6.05 | 3.80 |
| POGDIFF (OURS) | **9.10** | **8.40** | **8.80** | **8.20** | **8.50** | **8.00** | **7.90** | **9.60** |

Table 11: **Recall for the AgeDB-IT2I Dataset.** See the detailed definition of recall in Appendix E.

| Datasets | AgeDB-IT2I | | | | | | VGGFace-IT2I | |
|---|---|---|---|---|---|---|---|---|
| Size | Small | | Medium | | Large | | Small | |
| Metric | Recall Score ↑ | | | | | | | |
| Shot | All | Few | All | Few | All | Few | All | Few |
| VANILLA | 0.017 | 0.000 | 0.104 | 0.167 | 0.196 | 0.200 | 0.033 | 0.000 |
| CBDM | 0.267 | 0.000 | 0.159 | 0.083 | 0.138 | 0.100 | 0.233 | 0.000 |
| T2H | 0.017 | 0.000 | 0.104 | 0.167 | 0.196 | 0.200 | 0.033 | 0.000 |
| POGDIFF (OURS) | **0.800** | **1.000** | **0.517** | **0.642** | **0.435** | **0.540** | **0.767** | **1.000** |

