# OpenReview forum: "PoGDiff: Product-of-Gaussians Diffusion Models for Imbalanced Text-to-Image Generation"
_ICLR.cc/2025/Conference — Submitted to ICLR 2025_

### Official Review · Reviewer_ZUxR · 2024-10-29

**Soundness:** 3
**Presentation:** 2
**Contribution:** 3
**Rating:** 6
**Confidence:** 4

**Summary:**

The paper proposes a novel method PoGDiff to address the long-tailed data distributions cause by the imbalanced datasets. This paper proposes a general fine-tuning approach, replacing the ground-truth distribution with a Product of Gaussians conditioned on a neighboring text embedding. Experiments are conducted on AgeDB-IT2I and DigiFace-IT2I using FID, DINO, Human Score and GPT-4o Evaluation evaluation metrics.

**Strengths:**

1.	Selecting neighboring embeddings from other samples is an interesting approach, as it can help get a new density.
2.	Figure 3 is interesting, but it would benefit from a more detailed explanation.

**Weaknesses:**

1.	“Encouraging the model to generate the same image given similar text prompts” may result in a loss of diversity in the generated images. How can this drawback be overcome?
2.	The paper mentions that in diffusion models, a data point is affected only by its text embedding. However, even with the same text embedding, different latent codes can produce images of varying quality. Additionally, classifier-free guidance and negative prompts also influence image generation.
3.	Why is directly smoothing the text embedding not feasible?
4.	What is the basis for hypothetically defining $\sigma_{y'}^2 = \frac{\sigma_{t}^2}{\psi[(x,y), (x',y')]}$ ?
5.	What does ‘Cat’ refer to in line 249? It doesn’t seem to be explained in the paper.  The author should define or explain this term when it's first introduced
6.	What does the superscript of s in Equation 9 represent? The previous definition of s did not include a superscript (e.g., Equation 8).

**Questions:**

1.	Artifacts from PoGDiff appear to be present in the images generated at low density (e.g., Figure 1, lower left corner, J. Willard Marriott), but not in those generated at high density. Is this a result of the model's limitations?
2.	The paper mentions that when training a diffusion model on an imbalanced dataset, existing models often struggle to generate accurate images for less frequent individuals. Personalized methods (e.g., CustomDiffusion, PhotoMaker) can use 3 to 5 images to learn an identity and generate accurate images for these less frequent individuals. What is the difference between PoGDiff and personalized methods that learn a specific identity?
CustomDiffusion: Multi-Concept Customization of Text-to-Image Diffusion
PhotoMaker: Customizing Realistic Human Photos via Stacked ID Embedding
3.	How to obtain the ground-truth distribution $q(x_{t-1}|x_t, x_0,y)$, when given $x_t$, $x_0$, and $y$ ?
4.	The $y^{'}$ in line 167 and the $y^{'}$in line 169 should be the same symbol.
5.	Fig. 3 is interesting, but the type and amount of data used in Fig. 3 is quite confusing to me.
6.	Equation 7 neglects $y^{'}$.
7.	Does the distance between the current text embedding $y$ and the sampled $y^{'}$ significantly affect the final generated results?

---

> ### Author Response · Authors · 2024-11-24
> **[1/3] Thank you for the encouraging and constructive comments**
>
> Thank you for your encouraging and constructive comments. We are glad that you found our method ``"novel"`` and our experiments ``"interesting"``. Below, we address your questions one by one in detail. We have also **included all discussions below in our revision** (with the changed part marked in blue).
>
> **Q1: "Encouraging the model to generate the same image given similar text prompts” may result in a loss of diversity in the generated images. How can this drawback be overcome?"**
>
> We apologize for the confusion. One of our primary objectives is to generate accurate images of the same individual while ensuring facial consistency. Therefore **diversity can be harmful**. For example, given a text input of "Einstein", generated images with high diversity would generate both male and females images; **this is obviously incorrect**. Therefore it is important to strike a balance between **diversity** and **accuracy**, a goal that our PoGDiff achieves.
>
> In order to clarify the question in **Figure 5** you mentioned, we have added a new figure, **Figure 6 of Appendix C.2** (which contains images from Column 1, 2, and 6 for each method in Figure 5) to provide a clearer comparison with the training images. Specifically, in **Figure 6 of Appendix C.2**:
>
> + **Ground-Truth (GT) Images**: We show the ground-truth images on the right-most 3 columns.
>
> + **Column 1 and 2 of SDv1.5, CBDM, PoGDiff, and GT**: In these cases, the **training** dataset contains **only two images per person**. With such limited data, it is impossible to introduce meaningful diversity.
>   + SDv1.5 fails to generate accurate images altogether in this scenario.
>   + While CBDM might appear to produce the "diversity" you mentioned, it does so incorrectly, as it generates an image of a woman when the target is Einstein (we circled those wrong samples in first column in **Figure 6 of Appendix**).
>   + In contrast, our PoGDiff can successfully generate accurate images (e.g., Einstein images in Column 1) while still enjoying sufficient diversity.
>
> + **Column 3 of of SDv1.5, CBDM, PoGDiff, and GT**: In this case, the training dataset includes around 30 images per person.
>   + SDv1.5 generates accurate images but with nearly identical expressions, offering minimal diversity.
>   + CBDM still fails to generate accurate depictions of the individual.
>   + In contrast, our PoGDiff successfully generates accurate images while introducing notable diversity.
>
> In summary, typical diversity evaluation in diffusion model evaluations, such as generating multiple types of trees for a "tree" prompt, is **not the focus of our setting** and may even be **misleading**. In our setting, the key is to balance accuracy and diversity.
>
> We have incorporated this discussion into the revised version of the paper and explicitly emphasize the problem settings to avoid any further confusion.
>
> We believe this addition will provide you (and other readers) with a better understanding and context for interpreting Figure 5. Feel free to let us know if you have any follow-up questions, which we are more than happy to answer.

---

> ### Author Response · Authors · 2024-11-24
> **[2/3] Thank you for the encouraging and constructive comments**
>
> **Q2: "The paper mentions that in diffusion models, a data point is affected only by its text embedding. However, even with the same text embedding, different latent codes can produce images of varying quality. Additionally, classifier-free guidance and negative prompts also influence image generation."**
>
> This is an excellent point. We agree with your observation that "even with the same text embedding, different latent codes can produce images of varying quality, and classifier-free guidance and negative prompts also influence image generation." We have revised the paper accordingly.
>
> The purpose of our Figure 3 was to compare diffusion models and our PoGDiff in a simplified example. We agree that in practice the image also depends on the random latent codes. Note that this is already taken care of by our PoGDiff model's probabilistic formulation.
>
> To clarify this further, we have also updated the description in Figure 3 in the revised version to explicitly refer to **conditional text-to-image diffusion models**.
>
> Additionally, we note that our experimental settings intentionally ignore negative prompts and other techniques so that we have a clean evaluation setting. These are orthogonal to our method.
>
> **Q3: "Why is directly smoothing the text embedding not feasible?"**
>
> This is an excellent question. Preliminary results indicate that directly smoothing the text embeddings does not yield meaningful improvements. Below we provide some insights into why this approach might fail. Suppose we have a text embedding $y$ and its corresponding neighboring embedding $y'$. Depending on their relationship, we are likely to encounter three cases:
>
> 1. **Case 1: $y' = y$.**
>    In this case, applying a reweighting method such as a linear combination results in no meaningful change, as the smoothing outcome is still $y$.
>
> 2. **Case 2: $y'$ is far from $y$.**
>    If $y'$ is significantly distant from $y$, combining them becomes irrelevant and nonsensical, as $y'$ no longer represents useful neighboring information.
>
> 3. **Case 3: $y'$ is very close to $y$.**
>    When $y'$ is close to $y$, the reweighting can be approximated as: $\alpha y + (1-\alpha) y' \approx y + (1-\alpha)(y' - y)$.
>    Since $y'$ is nearly identical to $y$, this effectively introduces a small weighted noise term $(1-\alpha)(y' - y)$ into $y$. In our preliminary experiments, this additional noise degraded the performance compared to the original baseline results.
>
> Based on these observations, direct smoothing of text embeddings appears ineffective and may even harm performance in some cases.
>
> **Q4: "What is the basis for hypothetically defining \sigma_{y'}^{2}"**
>
> Thank you for mentioning this. In Eq.(5), there is a coefficient $\frac{\lambda_{y'}}{\lambda_{t}} = \frac{\sigma_{t}^{2}}{\sigma_{y'}^{2}}$. By setting $\sigma_{y'}^{2} = \frac{\sigma_{t}^{2}}{\psi (\cdot)}$, the term $\sigma_{t}^{2}$ cancels out, effectively removing the timestep dependency. This approach is consistent with the DDPM paper [2].
>
> We have included this discussion in the revised version of the paper.
>
> **Q5: "What does ‘Cat’ refer to in line 249? It doesn’t seem to be explained in the paper. The author should define or explain this term when it's first introduced"**
>
> We apologize for the confusion. The term "Cat" refers to a "Categorical" distribution. For example, $Cat([0.2, 0.5, 0.3])$ represents a three-dimensional categorical distribution, where there is a 0.2 probability of selecting the first category, 0.5 probability of selecting the second, and 0.3 probability of selecting the third.
>
> **Q6: "What does the superscript of s in Equation 9 represent? The previous definition of s did not include a superscript (e.g., Equation 8)."**
>
> We apologize for the confusion. In Eqn. (9), $s$ represents the cosine similarity sampled by the weight of $\{w_j\}$ as defined in Eqn. (8). The superscript of $s$ in Eqn. (9) is further explained immediately after Eqn. (9). Specifically, $\psi_{img-sim}(x, x')$ denotes the image similarity between the original image $x$ and the sampled image $x'$. To ensure that the similarity measure is meaningful only when $x$ and $x'$ are of the same person, we introduced the superscript to end-to-end control the image similarity based on their cosine similarity.
>
> For example, if the cosine similarity ($s$) between $x$ and $x'$ is 0.4, and $a_1=a_2=1$:
> + If $x$ and $x'$ are of the same person, the image similarity will be $0.4^{1}$.
> + If $x$ and $x'$ are not of the same person, the image similarity will be $0.4^{2}$, which is smaller.
>
> We have added further details to the main paper to clarify this point.

---

> ### Author Response · Authors · 2024-11-24
> **[3/3] Thank you for the encouraging and constructive comments**
>
> **Q7: "Artifacts from PoGDiff appear to be present in the images generated at low density (e.g., Figure 1, lower left corner, J. Willard Marriott), but not in those generated at high density. Is this a result of the model's limitations?"**
>
> Thank you for pointing this out. This is not a limitation specific to our method; for example Figure 1 shows that the baseline Stable Diffusion also suffers from this issue. Addressing these artifacts at lower densities is an interesting direction for future work.
>
> **Q8: "The paper mentions that when training a diffusion model on an imbalanced dataset, existing models often struggle to generate accurate images for less frequent individuals. Personalized methods (e.g., CustomDiffusion, PhotoMaker) can use 3 to 5 images to learn an identity and generate accurate images for these less frequent individuals. What is the difference between PoGDiff and personalized methods that learn a specific identity? CustomDiffusion: Multi-Concept Customization of Text-to-Image Diffusion PhotoMaker: Customizing Realistic Human Photos via Stacked ID Embedding"**
>
> Thank you for pointing us to these interesting papers [3, 4]. We have cited and discussed them in the revision.
>
> We would also like to clarify that our paper focuses on a setting different from works like CustomDiffusion and PhotoMaker. We provide more details below.
>
> **Different Setting from Custom Techniques like CustomDiffusion [3] and PhotoMaker [4].** Previous works like CustomDiffusion and PhotoMaker focus on adjusting the model to generate images with **a single object**, e.g., a specific dog. In contrast, our PoGDiff focuses finetuning the diffusion model on an entire data with **many different objects/persons simultaneously**. They are **very different settings** and are **complementary** to each other.
>
> **Q9: "How to obtain the ground-truth distribution"**
>
> Thank you for your question. According to DDPM and related works, the ground-truth distribution can be computed as outlined in Eq. 7 of the DDPM paper.
>
> **Q10: "The y' in line 167 and the y' in line 169 should be the same symbol."**
>
> We are sorry for the confusion. We have fixed this typo in the revision.
>
> **Q11: "Fig. 3 is interesting, but the type and amount of data used in Fig. 3 is quite confusing to me."**
>
> Sorry for the confusion. We meant to say that $y$ represents the text prompts, which are the embeddings of the text descriptions of the images, while $x$ corresponds to the associated images. Additionally, the tightly packed circles at the top indicate higher density, whereas the sparse circles represent lower density. To improve clarity, we have added more detailed explanations to the legend in Fig. 3.
>
> **Q12: "Equation 7 neglects y'"**
>
> We apologize for the confusion. In Eq.7, our $\psi_{inv-txt-den}$ represents the inverse of text density of the original data point. Therefore it does *not* contain $y'$.
>
> **Q13: "Does the distance between the current text embedding y and the sampled y' significantly affect the final generated results?"**
>
> This is a great question. The distance does indeed impact the final generated results, which is why we introduced a more sophisticated approach for computing $\psi$ in Eqs. (7) and (12). These mechanisms ensure that data points with smaller distances are assigned higher effective weights.
>
> Defining more robust and theoretically grounded methods to explore the text embedding space is an interesting direction for future work.
>
> [1] Qin et al. Class-Balancing Diffusion Models. CVPR 2023.
>
> [2] Ho et al. Denoising Diffusion Probabilistic Models. NeurIPS 2020.
>
> [3] Kumari et al. Multi-Concept Customization of Text-to-Image Diffusion. CVPR 2023.
>
> [4] Li et al. Customizing Realistic Human Photos via Stacked ID Embedding. CVPR 2024.

---

> ### Comment · Reviewer_ZUxR · 2024-11-25
>
> Thank you for the authors’ response. Most of my concerns have been resolved, and I have a few other questions.
>
> Q1: Maintaining facial consistency and accuracy is beneficial for your task. However, I wonder if there is a lack of diversity in the background or the angle of the face. It would be better to provide metrics for ID consistency and image diversity (such as Recall, Density, or Coverage) to demonstrate a balance between diversity and accuracy in PoGDiff. These evaluations can be conducted after segmenting the face and background.
>
> Q11: How many text descriptions are used in Fig. 3 to obtain the statistical results?

---

> > ### Author Response · Authors · 2024-11-26
> > **[1/2] Thank you for the encouraging and constructive comments**
> >
> > Thank you again for your follow-up response and the insightful question.
> >
> > **Q1: ... However, I wonder if there is a lack of diversity in the background or the angle of the face. It would be better to provide metrics for ID consistency and image diversity (such as Recall, Density, or Coverage) to demonstrate a balance between diversity and accuracy ...**
> >
> > Thank you very much for your question regarding the metric for diversity. The evaluation metrics and visualizations in our current revision already address this:
> >
> > **FID Measures Both ID Consistency and Diversity.** We would like to clarify that our Fréchet Inception Distance (FID) is computed for each ID separately, and the final FID score in the tables (e.g., Table 1) is the average FID over all IDs. Therefore FID measures both ID consistency and diversity.
> >
> > To see why, note that the FID score measures the distance between two Gaussian distributions, where the *mean* of the Gaussian represents the *identity (ID)* and the *variance* represents the *diversity*. For example, the *mean* of the ground-truth distribution represents the embedding position of the ground-truth ID, while the *variance* of the ground-truth distribution represents the *diversity* of ground-truth images. Similarly, the *mean* of the generated-image distribution represents the embedding position of the generated-image ID, while the *variance* of the generated-image distribution represents the *diversity* of generated images. A lower FID score indicates that the generated-image distribution more closely matches the ground truth distribution **in terms of both ID and diversity**.
> >
> > **Results Related to Diversity.** In our current revision:
> >   - **PoGDiff's Superior FID Performance.** In **Table 1**, we demonstrate that PoGDiff achieves a lower FID score, particularly in few-shot regions (i.e., minorities). This suggests that the images generated by our method capture a broader range of variations present in the training dataset, such as **backgrounds or facial angles**, as you mentioned.
> >   - **PoGDiff's Visualization.** We would like to direct your attention to **Figure 6 in the Appendix**. For example, in the minority group:
> >     - For Einstein (Column 1 for each method), the training dataset includes two face angles and two hairstyles. Our generated results successfully cover these attributes.
> >     - For JW Marriott (Column 2 for each method), the training dataset has only one face angle. Correspondingly our results focus on generating subtle variations in facial expressions with only one angle, **as expected**.
> >     - For the majority group (Column 3 for each method), our results clearly show that the generated images cover a wider range of diversity while maintaining ID consistency.
> >
> > **Additional Experiments on Recall (a New Metric).** Following your suggestion, we also design a new metric, "recall".
> > + **Recall in the Context of Image Generation: "Correct Image" and "Covered Image".** For each generated image, we classify it as a "correct image" if its distance to at least one ground-truth (GT) image is below a predefined threshold. For instance, suppose we have two training-set images for Einstein, denoted as $x_1$ and $x_2$. A generated image $x_g$ is a "correct image" if the cosine similarity between $x_g$ and either $x_1$ or $x_2$ is above some threshold (e.g., we set to $0.9$ here). For example, if the cosine similarity $x_g$ and $x_1$ is larger than $0.9$, we say that $x_g$ is a "correct image", and that $x_1$ is a "covered image". Intuitively, a training-set image (e.g., $x_1$) is covered if a diffusion model is capable of generating a similar image.
> > + **Formal Definition for Recall.** Formally, for each model, we compute the **Recall** per ID as follows:
> >     $$
> >     \text{Recall} = \frac{1}{c} \sum_{i=1}^{c} \frac{\text{number of unique covered images for ID i}}{\text{number of images for ID i in the training dataset}}
> >     $$
> > where $c$ is the number of IDs in a training set.
> > + **Cosine Similarity between Images.** Note that in practice, we compute the cosine similarity between DINO embeddings of images rather than raw pixels.
> > + **Analysis**: This metric evaluates the generational diversity of a model. For example, if the training dataset contains two distinct images of Einstein, $x_1$ and $x_2$, and a model generates only images resembling $x_1$, the recall in this case would be $0.5$. While the model may achieve high accuracy in terms of facial identity (Table 3 & Table 4), it falls short in diversity because it fails to generate images resembling $x_2$. In contrast, if a model generates images that cover both $x_1$ and $x_2$ the recall for this ID will be $1$; for instance, if the model generates 10 images for Einstein, where 6 of them resemble $x_1$ and 4 of them resemble $x_2$, the recall would be $1$, indicating high diversity and coverage.

---

> > > ### Author Response · Authors · 2024-11-26
> > > **[2/2] Thank you for the encouraging and constructive comments**
> > >
> > > **Additional Results in Terms of Recall.** Table A.1-A.3 below show the recall for different methods on three datasets, AgeDB-IT2I-small, AgeDB-IT2I-medium, and AgeDB-IT2I-large. These results show that our PoGDiff achieves much higher recall compared to all baselines, demonstrating its impressive diversity.
> > >
> > > Table A.1: Recall for AgeDB-IT2I-small in terms of DINO embedding.
> > > | model | overall | few |
> > > | :---------: | :------: | :------: |
> > > |VANILLA| 0.0167 | 0.00 |
> > > |CBDM| 0.2667 | 0.00 |
> > > |T2H| 0.0167 | 0.00 |
> > > |PoGDiff (Ours)| **0.80** | **1.00** |
> > > |||
> > >
> > > Table A.2: Recall for AgeDB-IT2I-medium in terms of DINO embedding.
> > > | model | overall | few |
> > > | :---------: | :------: | :------: |
> > > |VANILLA| 0.1037 | 0.1667 |
> > > |CBDM| 0.1591 | 0.0833 |
> > > |T2H| 0.1037 | 0.1667 |
> > > |PoGDiff (Ours)| **0.5169** | **0.6417** |
> > > |||
> > >
> > > Table A.3: Recall for AgeDB-IT2I-large in terms of DINO embedding.
> > > | model | overall | few |
> > > | :---------: | :------: | :------: |
> > > |VANILLA| 0.1965 | 0.20 |
> > > |CBDM| 0.1382 | 0.10 |
> > > |T2H| 0.1965 | 0.20 |
> > > |PoGDiff (Ours)| **0.4346** | **0.54** |
> > > |||
> > >
> > > **Additional details for Table A.1:**
> > > - For AgeDB-IT2I-small, there are two IDs, one "majority" ID with $30$ images and one minority ID with $2$ images.
> > > - For **VANILLA** and **T2H**, the recall for the majority ID and the minority ID is $1/30$ and $0/2$, respectively. Therefore, the average recall score is $0.5 * 1/30 + 0.5 * 0/2 \approx 0.0167$.
> > > - For **CBDM**, the recall for the majority ID and the minority ID is $16/30$ and $0/2$, respectively. Therefore, the average recall score is $0.5 * 16/30 + 0.5 * 0/2 \approx 0.2667$.
> > > - For **PoGDiff (Ours)**, the recall for the majority ID and the minority ID is $18/30$ and $2/2$, respectively. Therefore, the average recall score is $0.5 * 18/30 + 0.5 * 2/2 = 0.8$.
> > >
> > > We have included all results and discussion above in the **Appendix E** of the revision, and combined Table A.1-3 into **Table 6 in the Appendix E**.
> > >
> > > **Q11: How many text descriptions are used in Fig. 3 to obtain the statistical results?**
> > >
> > > This is a good question. We suppose you have the questions on clarifying the mapping between y and x in the figure 3. The purpose of Figure 3 is to illustrate that in our method, during the denoising process, each image $x$ is not only guided by its original text prompt $y$ but is also influenced by its neighboring text prompt(s) $y'$. In this paper, we focus on the case of a single $y'$, but for each sample pair $x, y$, a potentially different neighboring $y'$ will be randomly sampled in each epoch.
> > >
> > > In addition, our approach can naturally be extended to multiple $y'$ prompts. This extension poses both challenges and opportunities, making it an interesting direction for future research.
> > >
> > > Last but not least, thank you for keeping the communication channel open, and we hope the discussion above is helpful in clarifying your further questions. As always, feel free to let us know if you have any further questions, which we will strive to answer before the deadline.

---

> > > > ### Comment · Reviewer_ZUxR · 2024-11-29
> > > >
> > > > Thank you for your response. Most of my concerns and questions have been addressed. I still have a question about Figure 3. I believe that Figure 3 is an important observation and support for the paper, and the New Density can demonstrate the effectiveness of the proposed method.
> > > >
> > > > Regarding Figure 3 (left), y represents the text embedding, and x represents the corresponding generated image, as described in Line 181. The red dashed line indicates the Effective Density of the generated images. In my understanding, this Effective Density (represented by the red dashed line) should be derived from the statistical results of a large number of generated images. Therefore, I would like to know the amount of data, specifically the number of text (y) and the number of generated images (x), that are used to produce this statistical result (the red dashed line).

---

> > > > > ### Author Response · Authors · 2024-11-29
> > > > > **Thank you for the encouraging and constructive comments**
> > > > >
> > > > > Thank you for your follow-up question. We are glad that most of your concerns and questions have been addressed.
> > > > >
> > > > > For the question on Figure 3, to compute such effective density, we use 985 text-image pairs. Since our PoGDiff involves randomly sampling a neighboring input description for each text-image pair, effectively we have 1970 text-image pairs per epoch. Using 10 epochs, we have 19700 text-image pairs to compute the density. Note that for each epoch, different neighbors may be sampled.

---

> > > > > > ### Comment · Reviewer_ZUxR · 2024-11-30
> > > > > >
> > > > > > Thank you for your response. I do not have any further questions and will maintain my original rating.

---

> > > > > > > ### Author Response · Authors · 2024-11-30
> > > > > > > **Thank You**
> > > > > > >
> > > > > > > Dear Reviewer ZUxR,
> > > > > > >
> > > > > > > Thank you once again for your encouraging and valuable feedback. We are glad that our response has addressed all your concerns. We would be grateful if you might consider adjusting the score to reflect your current evaluation.
> > > > > > >
> > > > > > > Best regards,
> > > > > > >
> > > > > > > PoGDiff Authors

---

### Official Review · Reviewer_zjVi · 2024-10-30

**Soundness:** 2
**Presentation:** 2
**Contribution:** 2
**Rating:** 5
**Confidence:** 3

**Summary:**

This paper aims to enhance the performance of diffusion models when trained or fine-tuned on imbalanced datasets. Instead of relying on a single prompt, the authors align one image with multiple group prompts sampled from the training data. To achieve this, they employ a Product of Gaussians technique. The authors conduct various experiments to demonstrate that their method is effective.

**Strengths:**

1. The authors introduce the Product of Gaussians technique to sample mixed prompts, which alleviates the need for numerous small image-text pairs in the training dataset. This technique allows one image to interact with more prompts, thereby increasing generative potential.

2. The approach of using multiple texts to represent a single image is a reasonable consideration.

3. The authors provide theoretical support for their proposed method, which is intriguing.

4. Extensive experiments are conducted to support this idea.

5. The paper presents valuable insights into the proposed methodology.

**Weaknesses:**

1. The authors consider the use of non-target prompts for the current image, which may introduce noise and misalignment. This could result in generated images that do not align well with the prompts, potentially leading to lower CLIP scores, a metric that is not reported in the paper. Thus, there may be issues with text-image alignment.

2. The baseline comparisons are limited, focusing only on SD and CBDM, which may not be sufficient to fully validate the proposed idea.

3. As shown in Table 1, the proposed method does not demonstrate a significant advantage compared to the baselines, leaving me unconvinced about its effectiveness.

4. The visualizations do not clearly support the proposed method. For instance, Figure 5 reveals a lack of diversity in the generated outputs.

**Questions:**

see Weaknesses

---

> ### Author Response · Authors · 2024-11-24
> **[1/4] Thank you for the encouraging and constructive comments**
>
> Thank you for your valuable comments. We are glad that you found our method ``"reasonable"``, our theoretical analysis ``"intriguing"``, and our experiments ``"extensive"``. Below, we address your questions one by one in detail. We have also **included all discussions below in our revision** (with the changed part marked in blue).
>
> **Q1: "The authors consider the use of non-target prompts for the current image, which may introduce noise and misalignment. This could result in generated images that do not align well with the prompts, potentially leading to lower CLIP scores, a metric that is not reported in the paper. Thus, there may be issues with text-image alignment."**
>
> This is a good question.
>
> **Our PoGDiff Effectively Prevents Misalignment.** Empirically we did not observe such misalignment issue in our PoGDiff. This is because
> + PoGDiff leverages neighboring prompts $y'$ with *larger importance weights* on *closer* neighbors using cosine similarity $s$ of their corresponding images.
> + PoGDiff exponentially downweights unrelated prompts. For example, with $s\in [0,1]$, we use $s$ for similar prompts and $s^3$ for unrelated prompts, as shown in Eqn. (9) of the paper.
> + These neighboring prompts $y'$ are also weighted by their probability density, approximated by $ELBO_{VAE}(y')$, as shown in Eqn. (10) of the paper. This also effectively downweights less common or outlier neighboring prompts, preventing misalignment.
> + Our product-of-Gaussian training objective also helps prevent misalignment due to the effect of less similar prompts.
>
> In contrast, our baseline method CBDM [1] severely suffers from misalignment. Specifically, CBDM randomly samples prompts from the prompt space without any restrictions and pair them with the original images during training. This can lead to misalignment issues that you mentioned, as shown in our empirical results in **Table 1-4 and Figure 5**.
>
> **CLIP Scores Are Not Applicable in Our Setting.**
> Note that the CLIP score is not applicable in our setting. Specifically, our text prompts are predominantly human names. However, CLIP is primarily trained on common objects, not human names; therefore the CLIP score can not be use to compute matching scores between images and human names.
>
> **FID, Human Score, and GPT-4o Score Already Evaluates Alignment.** We would also like to clarify that our FID, Human Score, and GPT-4o Score already effectively evaluate the alignment between text prompt and the generated images.
> + Note that our FID is per-person FID. Specifically, for each person in the dataset, we compute the FID between the generated images and the corresponding real images. We can use the average FID across all persons as the final FID in Table 1. Therefore, for a given person, lower FID indicates that our generated face images align better with the ground-truth face images.
> + For Human Score and GPT-4o Score, humans and GPT-4o are directly queried to measure the alignment between the text prompt and the associated generated images. Therefore they also effectively evaluate text-image alignment.

---

> ### Author Response · Authors · 2024-11-24
> **[2/4] Thank you for the encouraging and constructive comments**
>
> **Q2: "The baseline comparisons are limited, focusing only on SD and CBDM, which may not be sufficient to fully validate the proposed idea."**
>
> Thank you for mentioning this. Actually, CBDM [1] is the most recently published work focusing on imbalanced learning in diffusion models.
>
> In addition, following your suggestion, we have included another similar work T2H [2] similar to CBDM [1] in our paper. Note that we did not include T2H [2] as a baseline in the original submission because it is not directly applicable to our setting. Specifically, T2H [2] relies on the class frequency, which is not available in our setting. Inspired by your comments, we adapted this method to our settings by using the density for each text prompt embedding to serve as the class frequency in T2H [2].
>
> We included the new results in **Table 1.1-4.3 below** and in **Table 1-4 of the main paper and Figure 7 of Appendix D**.
>
> Table 1.1: Results for accuracy in AgeDB-IT2I-small in terms of FID score.
> | model | overall | few |
> | :---------: | :------: | :------: |
> |VANILLA| 14.88 | 13.72 |
> |CBDM| 14.72 | 14.13 |
> |T2H| 14.85 | 13.66 |
> |PoGDiff (Ours)| **14.15** | **12.88** |
> |||
>
> Table 1.2: Results for accuracy in AgeDB-IT2I-medium in terms of FID score.
> | model | overall | few |
> | :---------: | :------: | :------: |
> |VANILLA| 12.87 | 12.56 |
> |CBDM| 11.63 | 11.59 |
> |T2H| 14.85 | 13.66 |
> |PoGDiff (Ours)| **14.15** | **12.88** |
> |||
>
> Table 1.3: Results for accuracy in AgeDB-IT2I-large in terms of FID score.
> | model | overall | few |
> | :---------: | :------: | :------: |
> |VANILLA| 7.67 | 11.67 |
> |CBDM| 7.18 | 11.12 |
> |T2H| 7.61 | 11.64 |
> |PoGDiff (Ours)| **6.03** | **10.16** |
> |||
>
> Table 1.4: Results for accuracy in Digiface-IT2I-large in terms of FID score.
> | model | overall | few |
> | :---------: | :------: | :------: |
> |VANILLA| 7.18 | 12.23 |
> |CBDM| 6.96 | 12.72 |
> |T2H| 7.14 | 12.22 |
> |PoGDiff (Ours)| **6.84** | **11.21** |
> |||
>
> Table 2.1: Results for accuracy in AgeDB-IT2I-small in terms of DINO score.
> | model | overall | few |
> | :---------: | :------: | :------: |
> |VANILLA| 0.42 | 0.37 |
> |CBDM| 0.54 | 0.09 |
> |T2H| 0.43 | 0.39 |
> |PoGDiff (Ours)| **0.77** | **0.73** |
> |||
>
> Table 2.2: Results for accuracy in AgeDB-IT2I-medium in terms of DINO score.
> | model | overall | few |
> | :---------: | :------: | :------: |
> |VANILLA| 0.39 | 0.28 |
> |CBDM| 0.38 | 0.11 |
> |T2H| 0.42 | 0.29 |
> |PoGDiff (Ours)| **0.69** | **0.56** |
> |||
>
> Table 2.3: Results for accuracy in AgeDB-IT2I-large in terms of DINO score.
> | model | overall | few |
> | :---------: | :------: | :------: |
> |VANILLA| 0.34 | 0.25 |
> |CBDM| 0.41 | 0.26 |
> |T2H| 0.37 | 0.26 |
> |PoGDiff (Ours)| **0.66** | **0.52** |
> |||
>
> Table 2.4: Results for accuracy in Digiface-IT2I-large in terms of DINO score.
> | model | overall | few |
> | :---------: | :------: | :------: |
> |VANILLA| 0.42 | 0.36 |
> |CBDM| 0.34 | 0.16 |
> |T2H| 0.44 | 0.36 |
> |PoGDiff (Ours)| **0.64** | **0.49** |
> |||
>
> Table 3.1: Results for accuracy in AgeDB-IT2I-small in terms of human evaluation.
> | model | overall | few |
> | :---------: | :------: | :------: |
> |VANILLA| 0.50 | 0.00 |
> |CBDM| 0.50 | 0.00 |
> |T2H| 0.50 | 0.00 |
> |PoGDiff (Ours)| **1.00** | **1.00** |
> |||
>
> Table 3.2: Results for accuracy in AgeDB-IT2I-medium in terms of human evaluation.
> | model | overall | few |
> | :---------: | :------: | :------: |
> |VANILLA| 0.66 | 0.32 |
> |CBDM| 0.44 | 0.08 |
> |T2H| 0.66 | 0.32 |
> |PoGDiff (Ours)| **0.96** | **0.92** |
> |||
>
> Table 3.3: Results for accuracy in AgeDB-IT2I-large in terms of human evaluation.
> | model | overall | few |
> | :---------: | :------: | :------: |
> |VANILLA| 0.60 | 0.20 |
> |CBDM| 0.56 | 0.12 |
> |T2H| 0.60 | 0.20 |
> |PoGDiff (Ours)| **0.84** | **0.68** |
> |||

---

> ### Author Response · Authors · 2024-11-24
> **[3/4] Thank you for the encouraging and constructive comments**
>
> Table 4.1: Results for accuracy in AgeDB-IT2I-small in terms of GPT-4o evaluation.
> | model | overall | few |
> | :---------: | :------: | :------: |
> |VANILLA| 5.20 | 3.20 |
> |CBDM| 4.50 | 1.10 |
> |T2H| 5.50 | 3.10 |
> |PoGDiff (Ours)| **7.47** | **9.51** |
> |||
>
> Table 4.2: Results for accuracy in AgeDB-IT2I-medium in terms of GPT-4o evaluation.
> | model | overall | few |
> | :---------: | :------: | :------: |
> |VANILLA| 4.30 | 2.90 |
> |CBDM| 1.30 | 1.00 |
> |T2H| 4.60 | 3.00 |
> |PoGDiff (Ours)| **8.80** | **8.20** |
> |||
>
> Table 4.3: Results for accuracy in AgeDB-IT2I-large in terms of GPT-4o evaluation.
> | model | overall | few |
> | :---------: | :------: | :------: |
> |VANILLA| 4.90 | 3.60 |
> |CBDM| 3.10 | 1.70 |
> |T2H| 4.70 | 3.90 |
> |PoGDiff (Ours)| **8.50** | **8.00** |
> |||
>
> These results show that T2H performs even worse than CBDM, with performance similar to fintuning a SD model.
>
> We will consider moving Figure 7 to the main paper, replacing Figure 5 in the revision, if you feel it is helpful.
>
> In conclusion, we can see that simple data re-weighting / re-sampling does not work well, and this is why more sophisticated methods like our PoGDiff is necessary. We hope our work can lay the foundation for more practical imbalanced text-to-image generation methods in the community.
>
> Besides T2H, if there is a specific method you would like us to compare with, we are very happy to cite, evaluate, and include it in the discussion section of our revision before the discussion period ends on Nov 26 AOE.
>
> **Q3: "As shown in Table 1, the proposed method does not demonstrate a significant advantage compared to the baselines, leaving me unconvinced about its effectiveness."**
>
> Thank you for your question.
>
> **Importance of Other Metrics Beyond FID.** It is important to note that the FID score measures only the distance between Gaussian distributions of ground-truth and generated images, relying solely on mean and variance. As a result, it does not fully capture the nuances of our task. This is why we include additional evaluation metrics such as DINO Score, Human Score, and GPT-4o Score, to comprehensively verify our method's superiority (as shown in Table 2-4). (For more details on the metrics, please refer to our **response to Q1** above.)
>
> **Additional Experiments: Limitation of FID.** In addition, we have added a figure showcasing a t-SNE visualization for a minority class as an example, as shown in **Figure 9 of Appendix C.5**, to further illustrate the limitation of FID we mentioned above. As shown in the figure:
> + There are two ground-truth IDs (i.e., two ground-truth individuals) in the training set.
> + Our PoGDiff can successfully generate images similar to these two ground-truth ID while maintaining diversity.
> + All baselines, including CBDM, fail to generate accurate images according to the ground-truth IDs. In fact most generated images from the baselines are similar to other IDs, i.e., generating the facial images of wrong individuals.
> These results show that:
> + Our PoGDIff significantly outperforms the baselines.
> + FID fails to capture such improvements because it depends only on the mean and variance of the distribution, losing a lot of information during evaluation.
>
> For DINO Score, Human Score, and GPT-4o Score, our method **significantly outperforms** all baselines. For example, in Table 4, our PoGDiff achieves an average GPT-4o Score of **above 8.00** while the baselines' average GPT-4o Scores are **below 4.50**. We can see similar large improvements from our method in Table 2 and 3.
>
> **Focusing on Few-Shot Generation.** Note that our focus is on the quality of imbalanced generation. Therefore we believe the improvements in few-shot generation are more relevant. For example, even in FID, our method can cut the FID (lower is better) from 14.13 to 12.88 in the AgeDB-IT2I-small dataset, a 8.8% improvement.

---

> > ### Author Response · Authors · 2024-11-24
> > **[4/4] Thank you for the encouraging and constructive comments**
> >
> > **Q4: "The visualizations do not clearly support the proposed method. For instance, Figure 5 reveals a lack of diversity in the generated outputs."**
> >
> > We apologize for the confusion. One of our primary objectives is to generate accurate images of the same individual while ensuring facial consistency. Therefore **diversity can be harmful**. For example, given a text input of "Einstein", generated images with high diversity would generate both male and females images; **this is obviously incorrect**. Therefore it is important to strike a balance between **diversity** and **accuracy**, a goal that our PoGDiff achieves.
> >
> > In order to clarify the question in **Figure 5** you mentioned, we have added a new figure, **Figure 6 of Appendix C.2** (which contains images from Column 1, 2, and 6 for each method in Figure 5) to provide a clearer comparison with the training images. Specifically, in **Figure 6 of Appendix C.2**:
> >
> > + **Ground-Truth (GT) Images**: We show the ground-truth images on the right-most 3 columns.
> >
> > + **Column 1 and 2 of SDv1.5, CBDM, PoGDiff, and GT**: In these cases, the **training** dataset contains **only two images per person**. With such limited data, it is impossible to introduce meaningful diversity.
> >   + SDv1.5 fails to generate accurate images altogether in this scenario.
> >   + While CBDM might appear to produce the "diversity" you mentioned, it does so incorrectly, as it generates an image of a woman when the target is Einstein (we circled those wrong samples in first column in **Figure 6 of Appendix**).
> >   + In contrast, our PoGDiff can successfully generate accurate images (e.g., Einstein images in Column 1) while still enjoying sufficient diversity.
> >
> > + **Column 3 of of SDv1.5, CBDM, PoGDiff, and GT**: In this case, the training dataset includes around 30 images per person.
> >   + SDv1.5 generates accurate images but with nearly identical expressions, offering minimal diversity.
> >   + CBDM still fails to generate accurate depictions of the individual.
> >   + In contrast, our PoGDiff successfully generates accurate images while introducing notable diversity.
> >
> > In summary, typical diversity evaluation in diffusion model evaluations, such as generating multiple types of trees for a "tree" prompt, is **not the focus of our setting** and may even be **misleading**. In our setting, the key is to balance accuracy and diversity.
> >
> > We have incorporated this discussion into the revised version of the paper and explicitly emphasize the problem settings to avoid any further confusion.
> >
> > We believe this addition will provide you (and other readers) with a better understanding and context for interpreting Figure 5. Feel free to let us know if you have any follow-up questions, which we are more than happy to answer.
> >
> > [1] Qin et al. Class-Balancing Diffusion Models. CVPR 2023.
> >
> > [2] Zhang et al. Long-tailed diffusion models with oriented calibration. ICLR 2024
> >
> > [3] Szegedy et al. Rethinking the Inception Architecture for Computer Vision. CVPR 2016

---

> ### Author Response · Authors · 2024-11-26
> **[1/2] Additional updates for Q4**
>
> Dear Reviewer zjVi,
>
> Inspired by comments from Reviewer ZUxR, we have run additional experiments with a new metric. The new results further demonstrate our PoGDiff's performance in terms of diversity.
>
> **Additional Experiments on Recall (a New Metric).** To better evaluate the superiority of our PoGDiff, we propose a new metric, "recall".
> + **Recall in the Context of Image Generation: "Correct Image" and "Covered Image".** For each generated image, we classify it as a "correct image" if its distance to at least one ground-truth (GT) image is below a predefined threshold. For instance, suppose we have two training-set images for Einstein, denoted as $x_1$ and $x_2$. A generated image $x_g$ is a "correct image" if the cosine similarity between $x_g$ and either $x_1$ or $x_2$ is above some threshold (e.g., we set to $0.9$ here). For example, if the cosine similarity $x_g$ and $x_1$ is larger than $0.9$, we say that $x_g$ is a "correct image", and that $x_1$ is a "covered image". Intuitively, a training-set image (e.g., $x_1$) is covered if a diffusion model is capable of generating a similar image.
> + **Formal Definition for Recall.** Formally, for each model, we compute the **Recall** per ID as follows:
>     $$
>     \text{Recall} = \frac{1}{c} \sum_{i=1}^{c} \frac{\text{number of unique covered images for ID i}}{\text{number of images for ID i in the training dataset}}
>     $$
> where $c$ is the number of IDs in a training set.
> + **Cosine Similarity between Images.** Note that in practice, we compute the cosine similarity between DINO embeddings of images rather than raw pixels.
> + **Analysis**: This metric evaluates the generational diversity of a model. For example, if the training dataset contains two distinct images of Einstein, $x_1$ and $x_2$, and a model generates only images resembling $x_1$, the recall in this case would be $0.5$. While the model may achieve high accuracy in terms of facial identity (Table 3 & Table 4), it falls short in diversity because it fails to generate images resembling $x_2$. In contrast, if a model generates images that cover both $x_1$ and $x_2$ the recall for this ID will be $1$; for instance, if the model generates 10 images for Einstein, where 6 of them resemble $x_1$ and 4 of them resemble $x_2$, the recall would be $1$, indicating high diversity and coverage.
>
> **Additional Results in Terms of Recall.** Table A.1-A.3 below show the recall for different methods on three datasets, AgeDB-IT2I-small, AgeDB-IT2I-medium, and AgeDB-IT2I-large. These results show that our PoGDiff achieves much higher recall compared to all baselines, demonstrating its impressive diversity.
>
> Table A.1: Recall for AgeDB-IT2I-small in terms of DINO embedding.
> | model | overall | few |
> | :---------: | :------: | :------: |
> |VANILLA| 0.0167 | 0.00 |
> |CBDM| 0.2667 | 0.00 |
> |T2H| 0.0167 | 0.00 |
> |PoGDiff (Ours)| **0.80** | **1.00** |
> |||
>
> Table A.2: Recall for AgeDB-IT2I-medium in terms of DINO embedding.
> | model | overall | few |
> | :---------: | :------: | :------: |
> |VANILLA| 0.1037 | 0.1667 |
> |CBDM| 0.1591 | 0.0833 |
> |T2H| 0.1037 | 0.1667 |
> |PoGDiff (Ours)| **0.5169** | **0.6417** |
> |||
>
> Table A.3: Recall for AgeDB-IT2I-large in terms of DINO embedding.
> | model | overall | few |
> | :---------: | :------: | :------: |
> |VANILLA| 0.1965 | 0.20 |
> |CBDM| 0.1382 | 0.10 |
> |T2H| 0.1965 | 0.20 |
> |PoGDiff (Ours)| **0.4346** | **0.54** |
> |||
>
> **Additional details for Table A.1:**
> - For AgeDB-IT2I-small, there are two IDs, one "majority" ID with $30$ images and one minority ID with $2$ images.
> - For **VANILLA** and **T2H**, the recall for the majority ID and the minority ID is $1/30$ and $0/2$, respectively. Therefore, the average recall score is $0.5 * 1/30 + 0.5 * 0/2 \approx 0.0167$.
> - For **CBDM**, the recall for the majority ID and the minority ID is $16/30$ and $0/2$, respectively. Therefore, the average recall score is $0.5 * 16/30 + 0.5 * 0/2 \approx 0.2667$.
> - For **PoGDiff (Ours)**, the recall for the majority ID and the minority ID is $18/30$ and $2/2$, respectively. Therefore, the average recall score is $0.5 * 18/30 + 0.5 * 2/2 = 0.8$.
>
> We have included all results and discussion above in the **Appendix E** of the revision, and combined Table A.1-3 into **Table 6 in the Appendix E**.
>
> These new results, along with our original **response to Q4**, verify the diversity of our PoGDiff.

---

> > ### Author Response · Authors · 2024-11-27
> > **[2/2] Additional updates for Q4**
> >
> > Additionally, we would also like to clarify that **our FID already measures diversity (along with ID consistency)** and that a lot of our results (in both the original and revised paper) do demonstrate the impressive diversity of our PoGDiff's generated images. Below we provide more details.
> >
> > **FID Measures Both ID Consistency and Diversity.** We would like to clarify that our Fréchet Inception Distance (FID) is computed for each ID separately, and the final FID score in the tables (e.g., Table 1) is the average FID over all IDs. Therefore FID measures both ID consistency and diversity.
> >
> > To see why, note that the FID score measures the distance between two Gaussian distributions, where the *mean* of the Gaussian represents the *identity (ID)* and the *variance* represents the *diversity*. For example, the *mean* of the ground-truth distribution represents the embedding position of the ground-truth ID, while the *variance* of the ground-truth distribution represents the *diversity* of ground-truth images. Similarly, the *mean* of the generated-image distribution represents the embedding position of the generated-image ID, while the *variance* of the generated-image distribution represents the *diversity* of generated images. A lower FID score indicates that the generated-image distribution more closely matches the ground truth distribution **in terms of both ID and diversity**.
> >
> > **Results Related to Diversity.** In our current revision:
> >   - **PoGDiff's Superior FID Performance.** In **Table 1**, we demonstrate that PoGDiff achieves a lower FID score, particularly in few-shot regions (i.e., minorities). This suggests that the images generated by our method capture a broader range of variations present in the training dataset, such as **backgrounds or facial angles**.
> >   - **PoGDiff's Visualization.** We would like to direct your attention to **Figure 6 in the Appendix**. For example, in the minority group:
> >     - For Einstein (Column 1 for each method), the training dataset includes two face angles and two hairstyles. Our generated results successfully cover these attributes.
> >     - For JW Marriott (Column 2 for each method), the training dataset has only one face angle. Correspondingly our results focus on generating subtle variations in facial expressions with only one angle, **as expected**.
> >     - For the majority group (Column 3 for each method), our results clearly show that the generated images cover a wider range of diversity while maintaining ID consistency.

---

> > > ### Author Response · Authors · 2024-11-29
> > > **Your Feedback Would Be Appreciated**
> > >
> > > Dear Reviewer zjVi,
> > >
> > > Thank you once again for your valuable comments. Your suggestions on clarifying our problem settings, evaluation metrics and baselines were very helpful. We are eager to know if our responses have adequately addressed your concerns.
> > >
> > > Due to the limited time for discussion, we look forward to receiving your feedback and hope for the opportunity to respond to any further questions you may have.
> > >
> > > Yours Sincerely,
> > >
> > > Authors of PoGDiff

---

> > > > ### Author Response · Authors · 2024-12-02
> > > > **Thank you for the constructive comments**
> > > >
> > > > Dear Reviewer zjVi,
> > > >
> > > > Thank vou for vour review during the discussion period.
> > > >
> > > > In response to your suggestions, we conducted an additional baseline (T2H) to compare PoGDiff's performance in **Table 1.1-4.3**. In addition, we explain for the reason that CLIP score is not applicable to our settings, and propose a new evaluation metric **recall** to evaluate the performance across PoGDiff and other baselines in **Table A.1-A.3**.
> > > >
> > > > With the ICLR Discussion Period concluding soon Dec. 2nd (AOE) for reiewers and Dec. 3rd (AOE) for authors, we kindly request your feedback on whether our responses address your concerns or if there are additional questions or suggestions you would like us to address.
> > > >
> > > > Thank you once again for your time!
> > > >
> > > > Yours Sincerely,
> > > >
> > > > Authors of PoGDiff

---

### Official Review · Reviewer_pyS7 · 2024-10-31

**Soundness:** 3
**Presentation:** 2
**Contribution:** 3
**Rating:** 6
**Confidence:** 4

**Summary:**

This paper presents Product-of-Gaussians Diffusion Models, a approach to fine-tuning diffusion models for imbalanced text-to-image generation. PoGDiff addresses the challenges of generating minority class images by using a product of Gaussians (PoG) to combine original target distributions with nearby text embeddings. Experimental results show that PoGDiff outperforms traditional methods like Stable Diffusion and Class Balancing Diffusion Model (CBDM) across various datasets, particularly enhancing generation for minority classes.

**Strengths:**

1. PoGDiff introduces the use of Gaussian products for fine-tuning in imbalanced datasets, an original approach that improves minority class image generation.

2. The paper provides theoretical analysis, showing that PoGDiff retains diffusion model properties while better representing minority classes.

**Weaknesses:**

1. Experiments are primarily conducted on AgeDB-IT2I and DigiFace-IT2I, which may not fully represent real-world, large-scale imbalanced datasets. Additional testing on broader datasets is necessary.

2. PoGDiff relies on neighboring samples for minority class improvement, which may be less effective in sparse data settings. There is a lack of discussion on how the model handles extremely sparse data.

**Questions:**

see weakness.

---

> ### Author Response · Authors · 2024-11-24
> **Thank you for the encouraging and constructive comments**
>
> Thank you for your encouraging and constructive comments. We are glad that you found our method ``"original"``, our writing ``"clear"``, and that our experiments show that our method ``"outperforms traditional methods"``. Below, we address your questions one by one in detail. We have also **included all discussions below in our revision** (with the changed part marked in blue).
>
> **Q1: "Experiments are primarily conducted on AgeDB-IT2I and DigiFace-IT2I, which may not fully represent real-world, large-scale imbalanced datasets. Additional testing on broader datasets is necessary."**
>
> This is a good suggestion. Following your suggestion, we are in the process of adding another dataset to this paper and hope to have some preliminary results ready before the discussion period ends (Nov 26 AOE).
>
> We would also like to note that our AgeDB-IT2I-small and AgeDB-IT2I-medium are actually sparse dataset, compared to a much denser version, AgeDB-IT2I-large. Along with DigiFace-IT2I, they cover different sparsity levels across two different data sources. Please see our **response to Q2** below for more details.
>
>
> **Q2: "PoGDiff relies on neighboring samples for minority class improvement, which may be less effective in sparse data settings. There is a lack of discussion on how the model handles extremely sparse data."**
>
> We apologize for the confusion. Our AgeDB-IT2M-small and AgeDB-IT2M-medium datasets are actually very sparse and are meant for evaluate the sparse data setting you mention.
>
> For example, the AgeDB-IT2M-small only contains images from 2 persons, it is therefore a very sparse data setting, compared to AgeDB-IT2M-large with images across 223 persons.
>
> We are sorry this was not clearly conveyed in Figure 4 or mentioned in the main text. To address this, we have added a bar plot in **Figure 8 of Appendix C.4** (the original Figure 4 in the main paper is a stacked plot) corresponding to Figure 4 to better illustrate the sparsity of these datasets and have included a corresponding note in the main paper. From **Figure 8**, We can see that the sparsity gradually increases from AgeDB-IT2M-large through AgeDB-IT2M-medium to AgeDB-IT2M-small.
>
> While sparse settings are not our primary focus, we agree that addressing imbalanced image generation in such setting is an interesting and valuable direction, and we have included a discussion about this in the limitations section of the paper.

---

> > ### Author Response · Authors · 2024-11-28
> > **Updates for an Additional Dataset**
> >
> > To address both of your comments, (1) ``"additional testing on broader datasets is necessary" and (2) ``"may be less effective in sparse data settings,"`` we have included an additional dataset, VGGFace, and run additional experiments on VGGFace.
> >
> > Specifically, we constructed a subset from VGGFace2 [1], named **VGGFace-IT2I-small**. This is a **sparse** dataset consisting of two individuals: the majority group contains $30$ images, while the minority group contains only $2$ images.
> >
> > The results shown in **Tables A.1–A.5** below demonstrate that our **PoGDiff** consistently outperform all baselines, highlighting its robustness and superior performance even on **imbalanced and sparse** datasets.
> >
> > Table A.1: FID score (lower is better) in VGGFace-IT2I-small.
> > | model | overall | few |
> > | :---------: | :------: | :------: |
> > |VANILLA| 14.18 | 12.73 |
> > |CBDM| 13.85 | 13.21 |
> > |T2H| 14.16 | 12.74 |
> > |PoGDiff (Ours)| **13.68** | **11.11** |
> > |||
> >
> > Table A.2: DINO score (higher is better) in VGGFace-IT2I-small.
> > | model | overall | few |
> > | :---------: | :------: | :------: |
> > |VANILLA| 0.49 | 0.36 |
> > |CBDM| 0.52 | 0.06 |
> > |T2H| 0.48 | 0.37 |
> > |PoGDiff (Ours)| **0.84** | **0.79** |
> > |||
> >
> > Table A.3: Human evaluation score (higher is better) in VGGFace-IT2I-small.
> > | model | overall | few |
> > | :---------: | :------: | :------: |
> > |VANILLA| 0.50 | 0.00 |
> > |CBDM| 0.50 | 0.00 |
> > |T2H| 0.50 | 0.00 |
> > |PoGDiff (Ours)| **1.00** | **1.00** |
> > |||
> >
> > Table A.4: GPT-4o evaluation score (higher is better) in VGGFace-IT2I-small.
> > | model | overall | few |
> > | :---------: | :------: | :------: |
> > |VANILLA| 6.00 | 3.60 |
> > |CBDM| 4.67 | 1.33 |
> > |T2H| 6.05 | 3.80 |
> > |PoGDiff (Ours)| **7.90** | **9.60** |
> > |||
> >
> > Table A.5: Recall (higher is better) for VGGFace-IT2I-small in terms of DINO embedding.
> > | model | overall | few |
> > | :---------: | :------: | :------: |
> > |VANILLA| 0.0333 | 0.00 |
> > |CBDM| 0.2333 | 0.00 |
> > |T2H| 0.0333 | 0.00 |
> > |PoGDiff (Ours)| **0.7667** | **1.00** |
> > |||
> >
> > We have also added these additional results to **Appendix F** in the revision as suggested.
> >
> > [1] Cao et al. Vggface2: A dataset for recognising faces across pose and age.

---

> > > ### Comment · Reviewer_pyS7 · 2024-11-28
> > >
> > > Thank you for the author's response and additional experiments, all my concerns have been resolved, and I will maintain my current positive score.

---

> > > > ### Author Response · Authors · 2024-11-28
> > > > **Thank you**
> > > >
> > > > Dear Reviewer pyS7,
> > > >
> > > > Thank you very much for your further feedback. We are glad that our response addressed all your concerns. If you find our response helpful, could you please consider raising the score to reflect your current evaluation?
> > > >
> > > > Thanks again!
> > > >
> > > > Best regards,
> > > >
> > > > Authors of PoGDiff

---

### Official Review · Reviewer_yW9M · 2024-11-05

**Soundness:** 2
**Presentation:** 3
**Contribution:** 2
**Rating:** 5
**Confidence:** 4

**Summary:**

This paper argues that current Diffusion models are trained on imbalanced datasets.
To solve this problem, they propose a fine-tuning framework, PoGDiff.
PoGDiff replaces the ground-truth distribution with a Product of Gaussians (PoG), which is constructed by combining the original ground-truth targets with the predicted distribution conditioned on a neighboring text embedding.
Experiments show that PoGDiff effectively addresses the imbalance problem in diffusion models, improving both generations' accuracy and quality.

**Strengths:**

This paper addresses the issue of imbalanced training data in diffusion models and proposes a novel fine-tuning method. The core idea is to modify the ground-truth image supervision signal during training by incorporating neighboring text embeddings.
1) The issue addressed in this paper is valuable and important.
2) The proposed solution appears to be reasonable.
3) The writing and analytical approach of the paper are clear.
4) The experiments also demonstrate the effectiveness of the proposed method.

**Weaknesses:**

Although the approach of PoGDiff is reasonable and effective, and I understand that the addition of text embeddings can increase the diversity of the supervision signal, I still have the following concerns: 1) From the results shown in Figure 1, some of the images generated by PoGDiff exhibit noticeable deviations in color and other aspects from the ground truth (GT). Does this modification align with the expected outcomes? 2) There are already several custom techniques that can achieve diversity with just a single or a few new style images, and in some cases, without any training. The fine-tuning method proposed by the authors might degrade the performance of the original model. How should we evaluate this? 3) The proposed method essentially resembles data re-weighting, yet the experiments lack comparisons and detailed analyses with similar methods.

**Questions:**

see weakness

**Details Of Ethics Concerns:**

use face images

---

> ### Author Response · Authors · 2024-11-24
> **[1/2] Thank you for the constructive comments**
>
> Thank you for your constructive comments. We are glad that you found the problem we addressed ``"valuable and important"``, our method ``"novel"``, our writing ``"clear"``, and that our experiments ``"demonstrate the effectiveness of the proposed method"``. Below, we address your questions one by one in detail. We have also **included all discussions below in our revision** (with the changed part marked in blue).
>
> **Q1: "From the results shown in Figure 1, some of the images generated by PoGDiff exhibit noticeable deviations in color and other aspects from the ground truth (GT). Does this modification align with the expected outcomes?"**
>
> This is a good question. We would like to clarify that color deviation is very common and is a known issue when one fine-tunes diffusion models (as also mentioned in [1]); for example, we can observe similar color deviation in both baselines (e.g., CBDM and Stable Diffusion v1.5) and our PoGDiff. This can be mitigated using the exponential moving average (EMA) technique [1]; however, this is orthogonal to our method and is outside the scope of our paper.
>
> We have included the discussion above in the revised paper.
>
> **Q2.1: "There are already several custom techniques that can achieve diversity with just a single or a few new style images, and in some cases, without any training. "**
>
> Thank you for mentioning this. We would like to clarify that
> + our paper focuses on a setting different from works like DreamBooth [2], and
> + our focus is not on diversity, but on finetuning a diffusion model on an imbalanced dataset.
> We provide more details below.
>
> **Different Setting from Custom Techniques like DreamBooth [2].** Previous works like DreamBooth focus on adjusting the model to generate images with **a single object**, e.g., a specific dog. In contrast, our PoGDiff focuses finetuning the diffusion model on an entire data with **many different objects/persons simultaneously**. They are **very different settings** and are **complementary** to each other.
>
> **Diversity.** Note that while our PoG can naturally generate images with diversity, diversity is actually **not** our focus. Our goal is to finetune a diffusion model on an imbalanced dataset. For example, PoGDiff can finetune a diffusion model on an imbalanced dataset of employee faces so that the diffusion model can generate new images that match each employee's identity. In this case, we are more interested in "faithfulness" rather than "diversity".
>
> **Q2.2: "The fine-tuning method proposed by the authors might degrade the performance of the original model. How should we evaluate this?"**
>
> This is a good suggestion. Note that our goal is to adapt the pretrained diffusion model to a specific dataset; therefore the evaluation should focus on the target dataset rather than the original dataset used during pretraining. For example, when a user finetunes a model on a dataset of employee faces, s/he is not interested in how well the fine-tuned model can generate images of "tables" and "chairs".
>
> We agree that evaluating the model's performance on the original dataset used during pretraining would be an intriguing direction for future work, but it is orthogonal to our proposed PoGDiff and out of the scope of our paper.
>
>
> **Q3: "The proposed method essentially resembles data re-weighting, yet the experiments lack comparisons and detailed analyses with similar methods."**
>
> Thank you for your question. Actually, one of our baselines, CBDM [3] can be considered as a data-reweighting method. As shown in **Figure 1 and Tables 1-4** in the paper, our PoGDiff can significantly outperform CBDM. This shows that simple data re-weighting does not work well, and therefore motivates our PoGDiff.
>
> There is another work T2H [4] similar to CBDM [3]; they are both equivalent to direct reweighting/resampling. We did not include T2H [4] as a baseline because it is not directly applicable to our setting. Specifically, T2H [4] relies on the class frequency, which is not available in our setting. Inspired by your comments, we adapted this method to our settings by using the density for each text prompt embedding to serve as the class frequency in T2H [4]. Results show that it performs even worse than CBDM.
>
> In conclusion, we can see that simple data re-weighting does not work well, and this is why more sophisticated methods like our PoGDiff is necessary. We hope our work can lay the foundation for more practical imbalanced text-to-image generation methods in the community.

---

> > ### Author Response · Authors · 2024-11-24
> > **[2/2] Thank you for the constructive comments**
> >
> > **Q4: "Details Of Ethics Concerns: use face images"**
> >
> > Thank you for raising the ethical concerns regarding the dataset we use. We would like to clarify that all the images are from **celebrities** and **publicly available**. Therefore there are no privacy concerns; in fact these datasets have been widely used within the research community.
> >
> > [1] Song et al. Score-based generative modeling through stochastic differential equations. ICLR 2021.
> >
> > [2] Ruiz et al. Dreambooth: Fine tuning text-to-image diffusion models for subject-driven generation. CVPR 2023
> >
> > [3] Qin et al. Class-Balancing Diffusion Models. CVPR 2023.
> >
> > [4] Zhang et al. Long-tailed diffusion models with oriented calibration. ICLR 2024

---

> > > ### Author Response · Authors · 2024-11-29
> > > **Your Feedback Would Be Appreciated**
> > >
> > > Dear Reviewer yW9M,
> > >
> > > Thank you once again for your valuable comments. Your suggestions on clarifying generation performance and color deviations were very helpful. We are eager to know if our responses have adequately addressed your concerns.
> > >
> > > Due to the limited time for discussion, we look forward to receiving your feedback and hope for the opportunity to respond to any further questions you may have.
> > >
> > > Yours Sincerely,
> > >
> > > Authors of PoGDiff

---

> ### Author Response · Authors · 2024-12-02
> **Thank you for the constructive comments**
>
> Dear Reviewer yW9M,
>
> In response to your suggestions, we address your concerns, including color deviation, our problem settings, and discussion with other re-weighting methods, specifically
>
> - Color deviation is very common and is a known issue when one fine-tunes diffusion models (as also mentioned in [1]);
>
> - Our paper focuses on a setting different from works like DreamBooth [2], and
> our focus is not on diversity, but on finetuning a diffusion model on an imbalanced dataset. We provide more details below.
>
> - Our goal is to adapt the pretrained diffusion model to a specific dataset.
>
> - Actually, one of our baselines, CBDM [3] can be considered as a data-reweighting method. As shown in **Figure 1 and Tables 1-4 in the paper**, our PoGDiff can significantly outperform CBDM. This shows that simple data re-weighting does not work well, and therefore motivates our PoGDiff. We also include one more dicussion T2H [4], and report the baseline performance in **Tables 1-4 in the paper**.
>
> We also address your **Ethics Concerns**: All the images are from **celebrities** and **publicly available**; therefore there are no privacy concerns; in fact these datasets have been widely used within the research community.
>
> With the ICLR Discussion Period concluding soon Dec. 2nd (AOE) for reiewers and Dec. 3rd (AOE) for authors, we kindly request your feedback on whether our responses address your concerns or if there are additional questions or suggestions you would like us to address.
>
> Thank you once again for your time!
>
> Yours Sincerely,
>
> Authors of PoGDiff
>
> [1] Song et al. Score-based generative modeling through stochastic differential equations. ICLR 2021.
>
> [2] Ruiz et al. Dreambooth: Fine tuning text-to-image diffusion models for subject-driven generation. CVPR 2023
>
> [3] Qin et al. Class-Balancing Diffusion Models. CVPR 2023.
>
> [4] Zhang et al. Long-tailed diffusion models with oriented calibration. ICLR 2024

---

### Author Response · Authors · 2024-11-24
**Global Response**

We thank all reviewers for their valuable comments.

We are glad that the reviewers found our method ``"novel"``/``"original"``/``"reasonable"``/``"interesting"`` (yW9M, pyS7, zjVi, ZUxR), the problem we addressed ``"valuable and important"`` (yW9M), our theoretical analysis ``"valuable"``/``"intriguing"`` (zjVi, yW9M), our writing ``"clear"`` (yW9M), and that our experiments ``"demonstrate the effectiveness of the proposed method"`` (yW9M), are ``"extensive"`` (zjVi), and show that our method ``"outperforms traditional methods"`` (pyS7).

Below we address the reviewers' questions one by one in detail. We have cited all related references and **included all discussions/results below in our revision** (with the changed part marked in blue).

---

### Meta-Review · Area_Chair_Libh · 2024-12-19

**Metareview:**

This paper introduces PoGDiff, a fine-tuning method for improving the performance of diffusion models on imbalanced datasets. By aligning each image with multiple group prompts using a Product of Gaussians approach, the method replaces the ground-truth distribution with one conditioned on neighboring text embeddings.

The introduction, related work, and technical sections of this manuscript all focus on addressing the general problem of imbalanced datasets. The proposed technique is sufficiently generic and can be applied to a wide range of scenarios, rather than being limited to the current experiments designed specifically for facial images. From this perspective, the current experimental setup restricts the applicability of the algorithm and limits the potential comparison methods and benchmark experiments for the proposed approach.

**Additional Comments On Reviewer Discussion:**

In addition to being limited to facial images, the reviewers have raised significant concerns regarding the evaluation of diversity. Although the authors provided some explanations—such as the notion that excessive diversity, for instance in generating images of Einstein, could compromise the authenticity of the generated images—this rationale may not hold in more general scenarios. For example, in natural image categories, if the training set contains relatively few dog images, the focus would shift to generating sufficiently diverse dog images, thereby weakening the emphasis on identity constraints. Addressing this could significantly enhance the practical applicability of the proposed algorithm.

The authors made an effort to include additional experiments during the rebuttal, which is appreciated. However, it is evident that the authors have recognized the current incompleteness of the manuscript. It would be more beneficial for the authors to take their time to carefully revise the paper, thoroughly incorporating the reviewers' comments into a comprehensive revision for a future submission.

---

### Decision · Program_Chairs · 2025-01-22

Reject